# GenSim: Generating Robotic Simulation Tasks via Large Language Models

**Lirui Wang[1], Yiyang Ling[2,3]\*, Zhecheng Yuan[4]\*, Mohit Shridhar[5], Chen Bao[6], Yuzhe Qin[2], Bailin Wang[1], Huazhe Xu[4], Xiaolong Wang[2]**
[1]MIT CSAIL, [2]UC San Diego, [3]Shanghai Jiao Tong University,
[4]Tsinghua University, [5]University of Washington, [6]CMU
https://liruiw.github.io/gensim

## Abstract

Collecting large amounts of real-world interaction data to train general robotic policies is often prohibitively expensive, thus motivating the use of simulation data. However, existing methods for data generation have generally focused on scene-level diversity (e.g., object instances and poses) rather than task-level diversity, due to the human effort required to come up with and verify novel tasks. This has made it challenging for policies trained on simulation data to demonstrate significant task-level generalization. In this paper, we propose to automatically generate rich simulation environments and expert demonstrations by exploiting a large language models' (LLM) grounding and coding ability. Our approach, dubbed GenSim, has two modes: goal-directed generation, wherein a target task is given to the LLM and the LLM proposes a task curriculum to solve the target task, and exploratory generation, wherein the LLM bootstraps from previous tasks and iteratively proposes novel tasks that would be helpful in solving more complex tasks. We use GPT4 to expand the existing benchmark by ten times to over 100 tasks, on which we conduct supervised finetuning and evaluate several LLMs including finetuned GPTs and Code Llama on code generation for robotic simulation tasks. Furthermore, we observe that LLMs-generated simulation programs can enhance task-level generalization significantly when used for multitask policy training. We further find that with minimal sim-to-real adaptation, the multitask policies pretrained on GPT4-generated simulation tasks exhibit stronger transfer to unseen long-horizon tasks in the real world and outperform baselines by 25%. [1]

## 1 Introduction

Achieving general-purpose robotic policies necessitates significant amounts of data, which is labor-intensive to collect in the real world. Although simulation provides an economical solution for generating diverse amounts of data at the scene level and instance level (Akkaya et al., 2019; Kaufmann et al., 2023; Fang et al., 2022; Deitke et al., 2022), increasing task-level diversity in simulation remains challenging due to the significant human effort required, especially for complex tasks. For example, creating new tasks involves specifying new asset relationships and task progression, as well as ensuring achievability and transferability to other contexts such as the real world. Due to the challenges, typical human-curated simulation benchmarks usually have only tens to hundreds of tasks (Zeng et al., 2021; Yu et al., 2020; James et al., 2020).

Recent years have witnessed significant progress in Large Language Models (LLMs) (OpenAI, 2023; Bubeck et al., 2023; Anil et al., 2023) in natural language processing and further in code generation (Chen et al., 2021; Rozière et al., 2023) for various tasks. In robotics, LLMs have been applied in multiple aspects, ranging from user interface (Ahn et al., 2022; Shridhar et al., 2022; Lynch et al., 2023; Driess et al., 2023), to task and motion planning (Lin et al., 2023; Huang et al., 2022), summarizing robot logs (Liu et al., 2023), and cost and reward designs (Yu et al., 2023b; Ha et al., 2023), revealing impressive capabilities on physical grounding and code generation. In this work, we take a step further to investigate whether LLMs can be used to create diverse simulation tasks, tapping further into these capabilities. Our LLM-based framework, GenSim, provides an

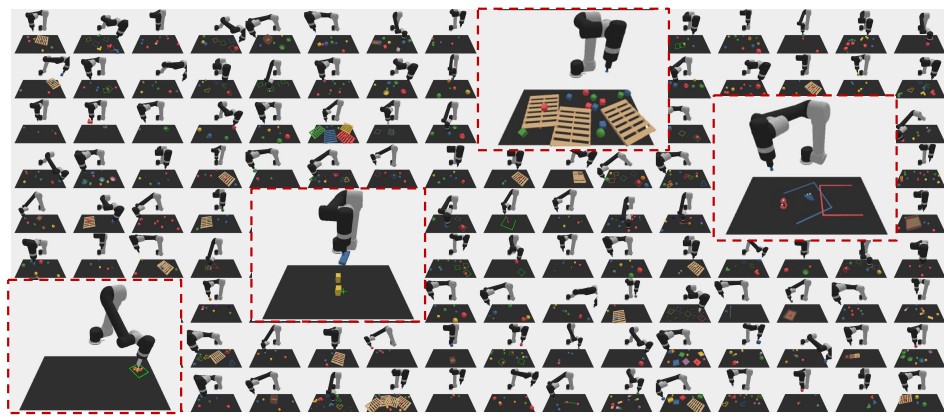

Figure 1: Task gallery of over 100 tasks generated by GPT4. GenSim leverages a LLM code generation pipeline to scale up simulation tasks for policy training and task-level generalization.

automatic mechanism to design and validate task asset arrangement and task progression. More importantly, the generated tasks exhibit great diversity, fostering task-level generalization of robotic policies. Conceptually in GenSim, the reasoning and coding capabilities of LLMs are distilled into lingo-visuo-action policies via the intermediately synthesized simulation data.

The framework is structured around three components: (1) *prompting mechanisms* that propose new tasks in natural language instruction and their corresponding implementations in code; this provides an automatic mechanism to design and validate task asset arrangement and task progression. (2) a *task library* that caches previously generated high-quality instruction code for validation and language model finetuning, returned as a comprehensive task dataset. (3) a *language-conditioned multi-task policy training procedure* that leverages the generated data to enhance task-level generalization. The framework operates in two distinct modes. In the goal-directed setting, where the user has a specific task or desires to design a task curriculum, the framework adopts a "top-down" approach. It takes the desired task as input and iteratively generates related tasks to attain the targeted objective. Conversely, in the exploratory setting, where no prior knowledge of the target task is available, the framework gradually explores beyond existing tasks and aims to establish a task-agnostic foundational policy.

By initializing the task library with 10 human-curated tasks (Shridhar et al., 2022) , we use GenSim to scale it up and generate *over 100* tasks (Figure 1). We propose several tailored metrics to progressively measure the quality of generated simulation tasks, and evaluate several LLMs in both goal-directed and exploratory settings. Based on the generated task library from GPT4, we conduct supervised finetuning on, including GPT3.5 and Code-Llama to further improve task generation performance of LLMs. Additionally, we quantitatively measure task achievability by policy training, and present task statistics across different properties, and code comparisons among different models.

Importantly, we have trained multitask robotic policies that generalize well on the generated tasks altogether and improve zero-shot generalization performance compared to models that train on only human-curated tasks. Furthermore, we also show that training jointly with GPT4-generated tasks can improve the generalization performance by 50% and have around 40% zero-shot transfer to new tasks in simulation. Finally, we consider sim-to-real transfer and show that pretraining on diverse simulation tasks achieves better real-world generalization capabilities by 25%. Overall, policies that trained on diverse LLM-generated tasks have better task-level generalization to new tasks, which indicates the potential of training foundational policies by scaling simulation tasks with LLM. The contributions of this work can be summarized as follows:

- We propose a novel simulation task generation pipeline through LLMs to generate over 100 tasks. We observe that LLMs are capable of generating high-quality, achievable, and diverse tasks by bootstrapping from existing human-curated tasks.

- We benchmark state-of-the-art LLM models such as GPTs and Code-Llama on simulated manipulation task creations. We find that prompting and finetuning based on the task library can significantly improve the capability of LLMs generating higher quality tasks.

---

[1] See our project website (https://liruiw.github.io/gensim), demo (https://huggingface.co/spaces/Gen-Sim/Gen-Sim), and code (https://github.com/liruiw/GenSim) for more details.

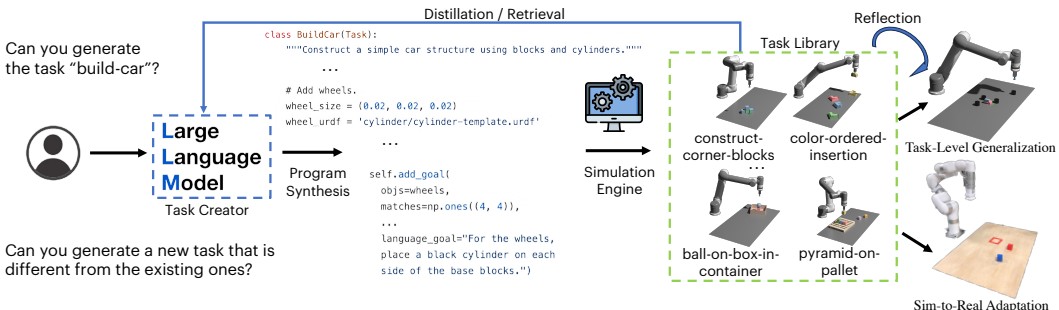

Figure 2: GenSim is an LLM framework to scale up simulation task diversity for robotic policy training. We investigate goal-directed mode (top prompt) and exploratory mode (bottom prompt) that generates robotic simulation task codes. The generated task codes are cached in a task library which can be used for policy training to achieve better task-level generalization and sim-to-real adaptations.

- We illustrate the effectiveness of the GPT4-generated tasks for language-conditioned visuo-motor policies by improving in-domain generalization by over 50% as well as zero-shot generalization to unseen tasks and instructions in both simulation and the real world.

## 2 AUTOMATED TASK GENERATION FOR POLICY LEARNING

We present GenSim (Figure 2), an LLM framework to generate simulation environments, tasks, and demonstrations through program synthesis. The GenSim pipeline starts with a task creator (Sec. 2.1) with a prompt chain operating in two modes, namely goal-directed mode and exploratory mode, depending on the knowledge of the target task. As a memory component, the task library (Sec. 2.2) is used to store previously generated high-quality tasks. Finally, the stored tasks from the library can be used for multitask policy training (Sec. 2.3) or finetuning LLMs for better task generation.

### 2.1 TASK CREATOR

The goal of the task creator is to propose novel task descriptions and corresponding code implementations, which can be further broken down into scene generations and demonstration generations. In particular, we use the Ravens benchmark (Zeng et al., 2021; Shridhar et al., 2022) which focuses on motion primitives such as pushing and pick-and-place that can be parameterized by two end-effector poses at each timestep. From the example in Figure 3, the reset function in the simulation environment code, efficiently initializes the assets and their attributes and poses, as well as the spatial and language goals that parameterize the action in each step[2]. In the exploratory task generation setting, the pipeline is prompted to generate a novel task that is sufficiently different from the existing tasks. In the goal-directed setting, the pipeline aims to fill in the task descriptions and implementations of a specified task name. The exploratory approach requires creativity and reasoning capability to come up with new tasks while goal-directed approach focuses on simulation coding as a specific task.

In both settings (Figure 3), the language chain first generates a task description and associated implementations afterward. The task description includes the task name, assets, and task summary. We adopt few-shot prompting for code generation in the pipeline. LLM is prompted to retrieve reference tasks and codes from existing tasks in the task library introduced in the next section. This process is critical for LLM to know exactly how to implement a task class (such as the procedure of sampling asset URDFs and building scene first, and then adding a spatial goal and language goals). In contrast to other LLM coding tasks, there are various feedback forms in robotic simulations including the execution pipeline, simulators, policy training, and humans. See Appendix §D for more details on the prompts and generated example code.

### 2.2 TASK LIBRARY

In the GenSim framework, we leverage an external memory, dubbed task library, to retrieve the generated tasks by the task creator to come up with better new tasks and to train multitask policies.

---

[2]Note that in this work, a task is defined by its code (and the associated language template) rather than a specific scene configuration, object relations, or demonstration trajectory.

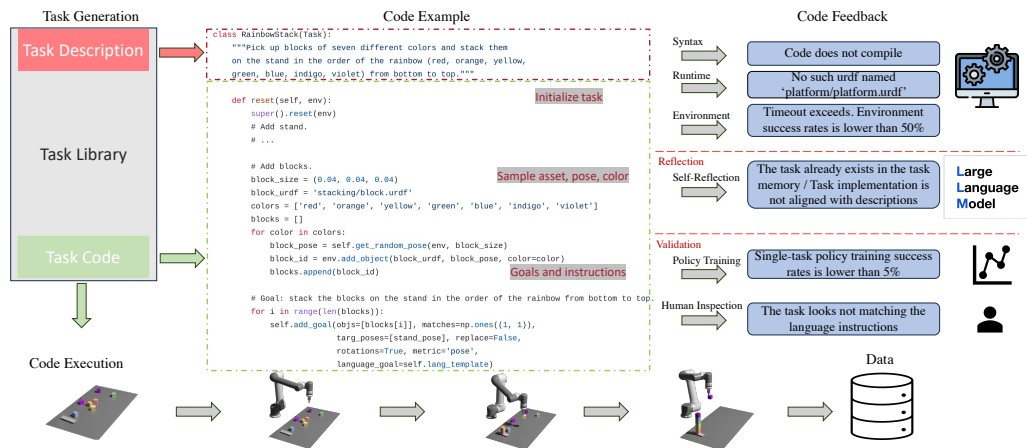

Figure 3: Our automatic simulation task generation pipeline (top left) generates a task code that can be used to generate scenes, simulations, and expert demonstrations for imitation learning. In addition to common execution-based feedback in LLM program synthesis tasks, the LLM critic and the task library provide task quality feedback. Finally, humans and single-policy training can provide the final on the expert and learner rollouts without any extensive coding experience (Appendix D.6)

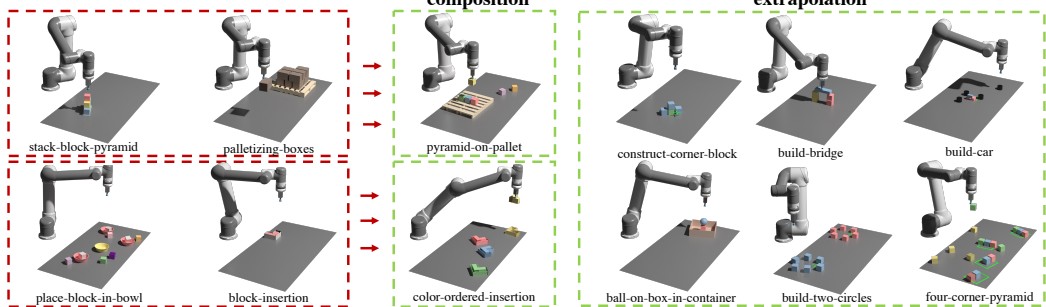

Figure 4: GenSim demonstrates interesting task-level composition and extrapolation behaviors in code generation for simulation tasks, which are distilled to policy learning through demonstrations.

The task library is initialized from the tasks in the human-curated benchmark. It provides the task creator with a list of past task descriptions to condition on in the description generation stage and a list of past codes in the code generation stage. Then, the task creator is prompted to retrieve reference tasks from the task library as examples for coding new tasks, i.e. retrieval augmented generation (RAG). After the task implementation is finished and can successfully generate demonstrations, we then prompt the LLM to reflect on the new task and the task library and form an ensembled decision for whether the newly generated task should be added to the library. In Figure 4, we observe interesting composition and extrapolation behaviors in the tasks generated by GenSim. These saved task codes can be used offline to generate demonstration trajectory data for multitask policy training.

The task library, generated in exploratory mode, can be used as bootstrapping data for iteratively training task creators to generate better simulation tasks in goal-directed mode. This is important to scale up task generations and incorporate human feedback as finetuned models are more economical to use as task creators. In the next section, we discuss how to distill the task-level generalizations in LLM code generation into policy learning, by expanding the corpus of the training tasks.

## 2.3 LLM Supervised Multitask Policy

Once the tasks are generated, we can use these task implementations to generate demonstration data and train manipulation policies. We use a similar two-stream transporter network architecture as in Shridhar et al. (2022) to parametrize the policy with affordance predictions. The code generation to language-conditioned behavior cloning process can be viewed as the distilling process from LLMs into the low-level control and affordance of robot policies. Treating the program as an efficient representation of the task and associated demonstration data (Figure 5), we can define the embedding space among tasks, whose distance metric is more robust to varying factors from perception such as object poses and shapes and yet more informative than the language instructions.

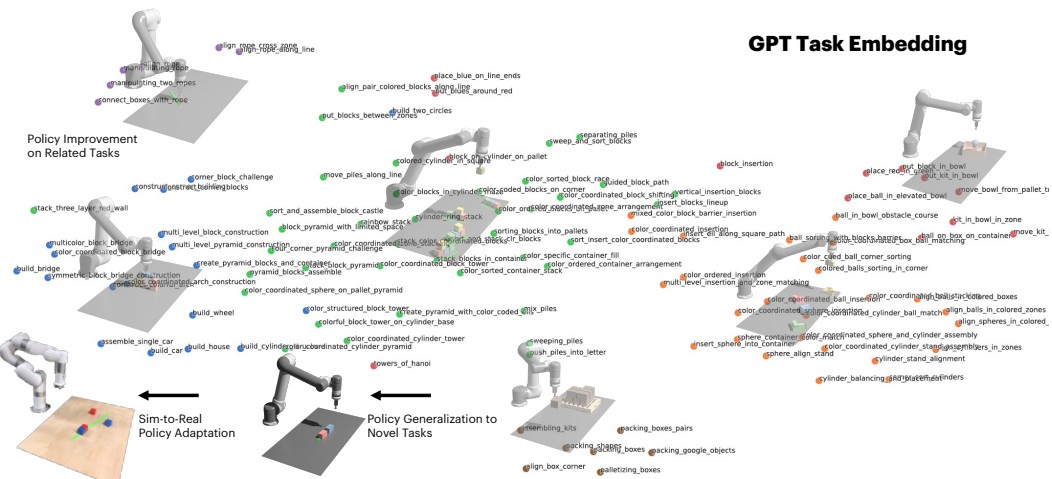

Figure 5: The task library can be used for retrieval and finetuning in GenSim Pipeline. Moreover, task code embedding can be used to create an embedding space in the task library (visualized as a T-SNE plot), which can be used for clustering tasks and policy training. For example, the purple represents the tasks involving rope, and blue denotes tasks that involve building structures.

## 3 EXPERIMENTS

In this section, we aim to use experiments to validate our framework, through the following specific questions: (1) How well can LLM design and implement simulation tasks? Can we improve the performance of LLMs on task generation? (2) Does training on LLM-generated tasks improve policy generalizations? Does policy training benefit more if more generated tasks are given? (3) Can pretraining on LLM-generated simulation tasks benefit real-world robot policy deployment?

### 3.1 EVALUATING LLM ROBOTIC SIMULATION TASK GENERATION

In this section, we ablated the design of the exploratory LLM pipeline with a simulation task-driven metric. Specifically, we measure a sequence of pass rates on "syntax-correct" which measures basic coding syntax issues as well as answer formatting problems, "runtime-verified" which tests asset hallucinations and code reasoning capability, and finally "task completed" which measures the designs of task demonstrations (the success in pick-place motions). These metrics are shown in the x-axis of Figure 6. Our metric has an incremental structure from syntax to runtime to successfully generate demonstrations where failing the former metrics implies failing on the latter metrics. Note that some problems such as misaligned language instruction and task behavior will not be captured by these metrics. To distill these task generation capabilities into more economic and scalable language models and to potentially conduct self-improvement, we use the 100 GPT4-generated tasks in the task library as a dataset for finetuning. We use OpenAI API to finetune GPT models and achieve better performance through this finetuning process. Moreover, we also finetune open-sourced LLMs such as Code-Llama (Rozière et al., 2023) with LoRA (Hu et al., 2021). We use the task names with a short prompt as input tokens and the task code as output tokens for autoregressive training.

**Exploratory task generation.** In total, the pipeline generates 120 diverse tasks that can be used for task demonstrations. We compare our task description and code separation prompt with a single prompt that requests both together as well as zero-shot prompt that does not provide as many reference codes. Figure 6 shows that a two-stage prompt chain with few-shot examples and task library can effectively improve the code generation success rates. The exploratory task generation requires the LLM to have language reasoning capabilities to understand the prompts and creativity for new tasks.

**Goal-directed task generation.** We also experiment with the goal-directed procedure to measure the coding capabilities of different language models. Specifically, we pick 10 held-out tasks and prompt each model to generate three trials (multiple implementations of the same task) for evaluating the metric. We observe that GPT4 is still outperforming other models in the specific robot simulation coding tasks. More specifically, strong closed-source models such as GPT-3.5 and GPT4 with in-context learning (prompting) can generate creative tasks and yet are still prone to hallucinations in code. We observe that the finetuned open-source models can achieve closer performance as state-of-the-art LLMs. The finetuned open-source models can generate the correct code flow and syntax but they occasionally exhibit misalignment between high-level objectives and implementation,

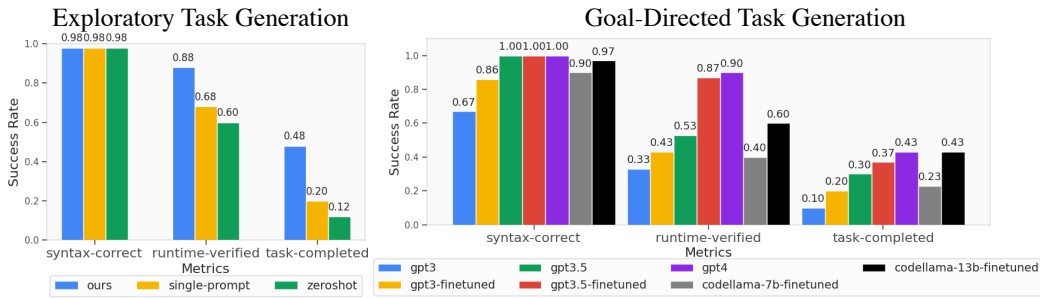

Figure 6: Left) The prompt chain with few-shot examples and the task library are helpful for LLM simulation task generation. Right) Finetuning on GPT4's generated tasks can improve simulation coding capabilities, for both closed-source GPT3.5 and open-source Code-LLama-Instruct-13B.

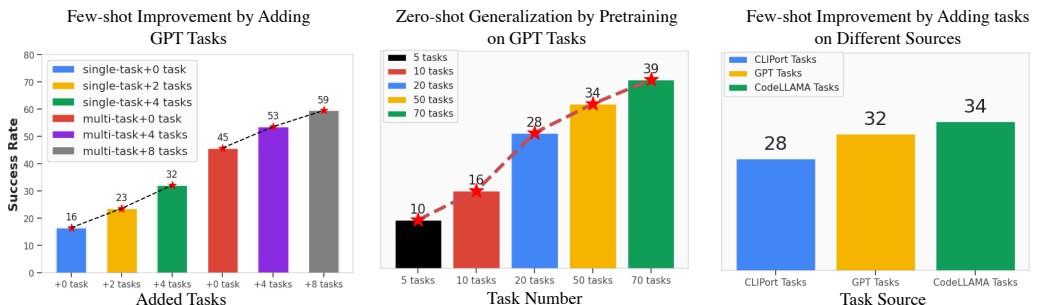

Figure 7: Joint training with augmented data generated from the GenSim tasks can improve policy performance, in both single-task and multi-task settings. Left) the x-axis denotes single CLIPort task + N GPT tasks where N=0,2,4 and multiple CLIPort tasks + N GPT tasks where N=0,4,8. Middle) We also showed that when training on more and more tasks, the policy exhibits stronger and stronger zero-shot generalization capabilities. Right) The performance of few-shot joint training varies for different sources of task generation including open-source LLMs.

due to the complexities of long simulation code. This motivates increasing better language models in simulation task creations such as self-instruct (Wang et al., 2022) as well as distillation from more capable models. See Appendix §A, §D for more experiment details and examples.

## 3.2 TASK-LEVEL GENERALIZATION

In this section, we study how the generated tasks from LLM can help with tabletop language-conditioned visuomotor policy learning for generalizations and adaptation. We adopt the 0 (fail) to 100 (success) scores proposed in the Ravens benchmark (Zeng et al., 2021) which considers partial credits for completing tasks. The simulation robot setup is a Universal Robot UR5e with a suction gripper. The policy input is a top-down RGB-D reconstruction, and the output is an affordance map that is then transformed to pick and place actions. We use the CLIPort (Shridhar et al., 2022) architecture but the framework is independent of which policy parametrization we use. The set of testing tasks is not particularly selected. We have released a language-conditioned benchmark ( with model weights and task lists) generated by GPT, ranging from 10 tasks to 100 tasks, to study scaling policy learning with affordance prediction.

**Few-shot policy generalization to related tasks**. In particular, from Fig. 7 left, we show that jointly training LLM-generated tasks can improve the policy performance on the original CLIPort (Shridhar et al., 2022) tasks by over 50%, especially under low data regime such as 5 demos. The result is averaged over two different task splits. This is expected as adding related tasks reduces the overfitting problem on a few demos. On the right, we also experimented with pretraining with different task sources of the human written tasks, closed-source LLM, and open-source finetuned LLM, and observed similar improved performance.

**Zero-shot policy generalization to unseen tasks**. From Fig. 7 middle, by pretraining on more tasks generated by LLM, our model can generalize better to tasks in the original Ravens benchmark. The task-level generalization is surprising to us considering that the task and language instructions have

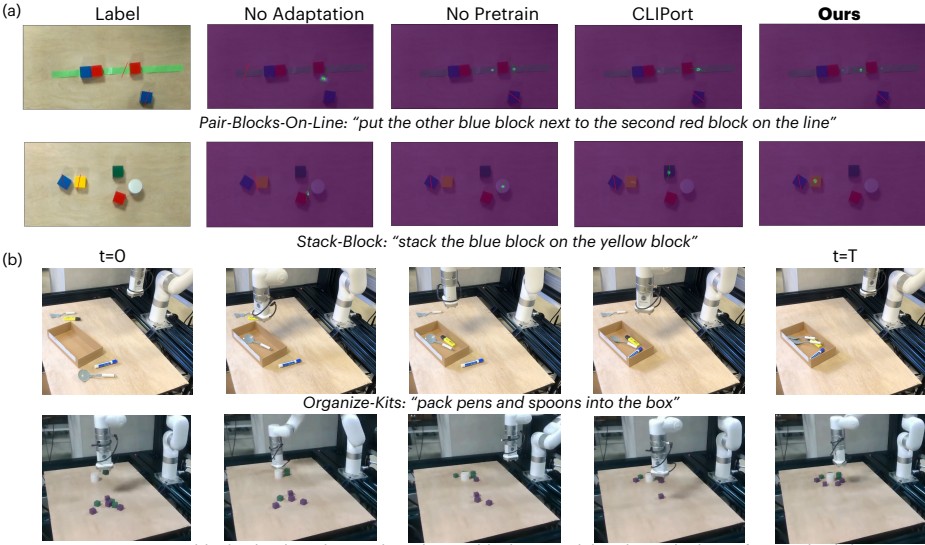

Figure 8: (a) The labels and the affordance heatmap of different real-world policies (green denotes placing and red denotes picking). (b) Executions of long-horizon tasks by GPT4-pretrained models.

| Task | Episodes(Sample) | No Adaptation Success | No Pretrain Success | CLIPort Success | GenSim (50 Tasks) Success | GenSim (70 Tasks) Success |
|---|---|---|---|---|---|---|
| put-block-in-bowl | 20(20) | 0/10 | 5/10 | 5/10 | 8/10 | 8/10 |
| stack-block | 25(25) | 0/10 | 2/10 | 6/10 | 9/10 | 10/10 |
| block-tower-on-corner | 20(80) | 0/10 | 0/10 | 2/10 | 3/10 | 4/10 |
| block-in-bowl-in-zone | 10(20) | 0/10 | 10/10 | 7/10 | 10/10 | 10/10 |
| put-block-at-zone-corner | 10(40) | 0/10 | 0/10 | 10/10 | 9/10 | 10/10 |
| pair-blocks-on-line | 20(80) | 0/10 | 0/10 | 0/10 | 2/10 | 4/10 |
| align-block-in-zone | 20(60) | 0/10 | 0/10 | 2/10 | 3/10 | 6/10 |
| build-wheel | 10(80) | 0/10 | 0/10 | 0/10 | 0/10 | 3/10 |
| pack-spheres | 20(80) | 0/10 | 6/10 | 7/10 | 4/10 | 7/10 |
| organize-kits | 15(60) | 0/10 | 5/10 | 5/10 | 3/10 | 6/10 |
| sort-garbage | 20(60) | 0/10 | 3/10 | 1/10 | 4/10 | 4/10 |
| place-bread | 15(60) | 0/10 | 2/10 | 2/10 | 1/10 | 3/10 |
| average | 205(665) | 0% | 27.5% | 39.2% | 46.7% | **62.5%** |

Table 1: Success rates (%) of multi-task policies that are finetuned on the base models from simulation. The test performance is measured across different scenes and tasks.

not been seen in the training datasets. Note that these tasks are selected using the distance metric on the task embedding (Figure 5). One intuition is that LLM code generation expands the task data, and thus training on these related tasks leads to more robust generalizable representations.

Specifically, as shown in Fig. 4, the *color-ordered-insertion* can be thought of as a compositional task between *block-insertion* and *place-block-in-bowl* from CLIPort original task tasks. The former task involves placing a block into some colored bowl, while the goal of the latter task is to insert a specific block into fixtures. In the left of Figure 7, our experiment suggests that learning jointly with related and more complex task *color-ordered-insertion* has the potential to contribute to enhanced generalization capabilities of *put-block-in-bowl*. On the other hand, when we try to learn a base policy for novel task generalization and even across domains, it is more beneficial to learn from as diverse tasks as possible, corresponding to exploratory task generation. See Appendix §B for more details.

## 3.3 ADAPTING PRETRAINED MODEL TO THE REAL WORLD

In this section, we conduct experiments to transfer the policy trained in simulation to the real environment. We hypothesize that by expanding the diversity of training tasks generated by LLM in the simulation, the trained policy would exhibit enhanced adaptability in real-world scenarios. To further enhance the sim-to-real transition, we incorporate an adaptation process for the real world. This process includes collecting a small set of real-world data for each task, followed by data augmentations and the fine-tuning of the simulation-pretrained model over 50 epochs. We perform our real-world experiments using an XArm-7 robot equipped with a suction gripper. A bird's-eye-view camera is installed facing downward to capture RGB-D observations. In Table 1, the model pretrained on 70 GPT4-generated tasks achieves an average success rate of 62.5% over 10 trials of 12 tasks, an over 20% increase compared to baselines that only pretrained on CLIPort tasks

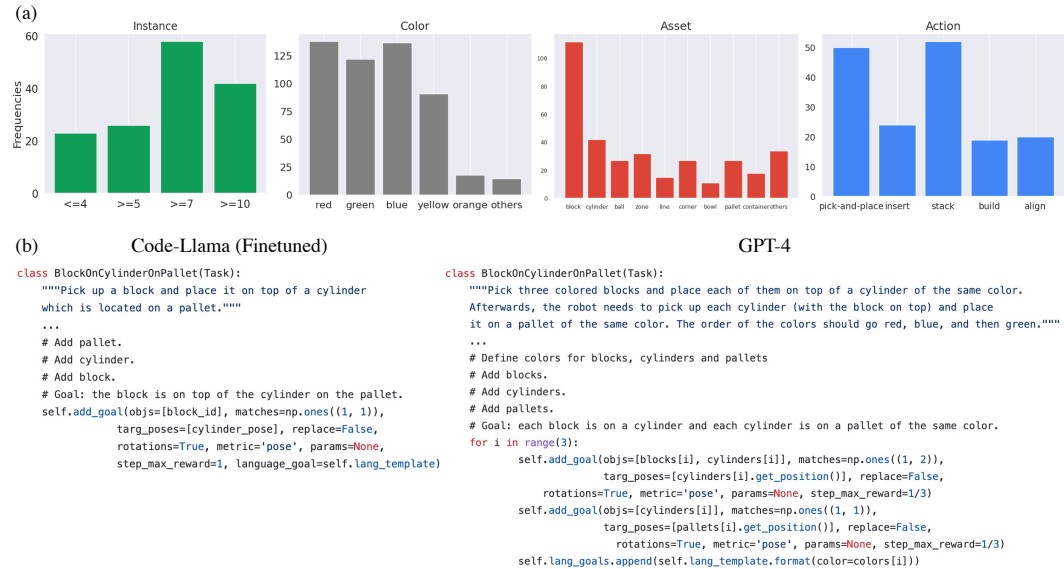

Figure 9: (a) Task statistics across different properties in the generated benchmark. (b) Example failure modes of the Code LLama after finetuning, and GPT4 through prompting.

and 15% improvement on models that pretrained on only 50 tasks. Qualitatively, baseline models without sufficient pretraining often pick or place the wrong objects, and baselines without adaptations have diffused affordance prediction. We hypothesize that GPT4-generated task pretraining exposes policy with broad language instructions and task compositions, which allows it to generalize better to a diverse set of tasks after sim-to-real adaptation. Moreover, we qualitatively observe that pretraining on diverse simulation tasks improves robustness on long-horizon complex tasks (Figure 8). For example, the GPT4-pretrained model has a more robust performance on *build-wheel* task in the real world. Additionally, we show that the simulation pretrained weights can also be useful for tasks such as place-bread and sort-garbage. These are distinct from the simulation tasks with different assets and semantic information. We defer the task, training, and evaluation details to the Appendix §C.

### 3.4 ADDITIONAL ANALYSIS

**Simulation Training Success Rates.** On Table 2, we showed single-task and multi-task policy training success rates on a subset of our generated tasks with 200 demos. The average task success rate for policy training on GPT4-generated tasks is *75.8%* for single-task and *74.1%* for multi-task, which is similar to *76.6%* for a single task and *76.1%* for multi-task from human-curated tasks in CLIPort (Shridhar et al., 2022). See Appendix §B for more training details.

**Generated Task Statistics.** In Figure 9 (a), we showed task statistics on different features of the 120 generated tasks by LLM. We observe an interesting balance of colors, assets, actions, and the number of instances generated by the LLM models. For instance, the generated code contains many scenes with over 7 object instances and many primitive actions of pick-and-place and assets such as blocks.

**Code Generation Comparison.** In Figure 9 (b), we qualitatively evaluate failure cases in the top-down experiments for both the Code Llama and GPT4. The coding objective is to implement the multi-step logic of placing a cylinder on a pallet first and then placing the block on the pallet. Based on the add goal step, we observed that Code LLama has ignored the pallet completely, and GPT4 got the order of placement wrong, while being a bit noisy in text descriptions. Other common LLM modes include misaligned language descriptions and task objectives as well as imbalanced task distribution when scaling the LLM generation pipeline. More details can be found in §A.5.

**Human Effort Evaluation.** In Section §D.6, we discuss and evaluate the human verification time spent and pass rates on the LLM-generated tasks. The average human time is around 10 seconds (if each task is checked) and the success rates are above 50%.

### 4 RELATED WORK

**Reasoning and Coding via LLMs.** There have been impressive emerging abilities of LLMs, including zero-shot prompting and intricate reasoning (Wei et al., 2022; Chowdhery et al., 2022) in the past few years. For example, Park et al. (2023); Zhang et al. (2023) use language models in

| Task | Single-Task | Multi-Task | | Task | Single-Task | Multi-Task |
|---|---|---|---|---|---|---|
| build-car | 84.1 | 85.4 | | insert-blocks-lineup | 84.1 | 80.0 |
| color-coordinated-sphere-insertion | 99.2 | 93.0 | | move-piles-along-line | 27.7 | 32.7 |
| cylinder-ring-stack | 82.0 | 81.0 | | manipulating-two-ropes | 30.8 | 40.2 |
| multi-level-block-construction | 67.8 | 49.0 | | color-ordered-insertion | 93.0 | 94.0 |
| stack-blocks-in-container | 99.0 | 98.8 | | align-cylinders-in-zones | 91.2 | 87.8 |

Table 2: Success rates (%) of the trained policy on example GPT4-generated tasks.

explorations of new tasks. Our work, which uses separate LLM critics to evaluate the generated programs and a memory for storing and reflecting previous outputs, is also related to the rich line of work on parameterizing agents with LLMs (Li et al., 2023; Wang et al., 2023). On the other hand, program synthesis has a long line of research in NLP (Manna & Waldinger, 1971; 1980; Chen et al., 2021; Wang et al., 2019; Chen et al., 2018). LLM models, such as Codex (Chen et al., 2021) make program synthesis accessible by only requiring a specification of docstring or incomplete code. Execution-based code generation (Ellis et al., 2019; Chen et al., 2018) uses execution outcomes at the code compiler level to iterate code generation. Skreta et al. (2023) uses a rule-based verifier to improve program generation and Wang et al. (2023) incorporates environment feedback and self-verification in Minecraft into the pipeline. Our work benchmarks simulation task creation with both open-source and closed-source LLM code models. In addition, our method also leverages multiple verification methods as feedback for simulation task scripts in robotic simulations.

**Task and Scene Generation for Simulation.** In robotic simulations, recent works have explored domain randomizations (Tobin et al., 2017; Fang et al., 2022; Chen et al., 2023; Ramos et al., 2019) and procedural asset generations (Deitke et al., 2022; Makatura et al., 2023) and text to 3D (Jun & Nichol, 2023; Nichol et al., 2022; Poole et al., 2022; Yu et al., 2023a). In this work, we treat each task as the code implementation and expand the original 10 tasks in the Ravens benchmark (Zeng et al., 2021; Shridhar et al., 2022) to over 100 novel tasks and associated expert demonstrations through LLM. Consequently, different from previous works that study compositional generalization (Jiang et al., 2022) or object-level (Shridhar et al., 2022) generalization, we study the task-level generalization of policy learning.

**Language Models in Robotics.** In robotics, large language models have been applied to policy learning (Driess et al., 2023), task and motion planning(Lin et al., 2023; Huang et al., 2022), summarizing logs (Liu et al., 2023), as well as synthesizing policy programs (Liang et al., 2022) and optimization programs (Yu et al., 2023b). Past work has also explored LLM's physical grounded capability (Liu et al., 2022) and concurrent work explores using LLM together with task and motion planners to create expert demonstrations (Ha et al., 2023). Ahn et al. (2022); Lynch et al. (2023) attempted to collect huge-scale real-world interactions but are only focused on specific task families. Instead of interacting in expensive real-world settings, we explored how to create increasingly complex simulation tasks and demonstrations jointly with LLM, and study the scaling law on policy generalizations through training on GPT-generated tasks.

## 5 CONCLUSION AND FUTURE WORKS

In this work, we present GenSim, a scalable LLM framework to augment diverse simulation tasks for robotic policy, which aims to distill LLM grounding and coding capabilities into low-level policies. We investigate LLM prompting, retrieval-augmented generation, and finetuning in both goal-directed and exploratory approaches to generate new simulation task codes. We leverage the generated tasks for training multitask policies that show generalization capabilities both to new tasks in simulation and the real world. We have seen some limitations in enabling LLM to design tasks and demonstrations in robotic simulation. The generated code still contains basic syntax errors and suffers from hallucinations and a lack of grounding in physical and geometric details. Another problem is that the code generation evaluation metric is imperfect (such as misaligned language description) and therefore the generated tasks can require some manual filtering before policy training. Finally, we have only explored table-top pick-and-place task generation, and generating dexterous and complex robotic tasks could be more challenging. We hope these limitations shed some light on future attempts at simulation code generations with LLMs.

## 6 ACKNOWLEDGEMENT

We thank MIT Supercloud for providing computing resources. The authors would like to thank many helpful discussions from Yoon Kim and Russ Tedrake at MIT. This work is supported in part by the Amazon Greater Boston Tech Initiative and Amazon PO No. 2D-06310236. This project was supported, in part, by the Amazon Research Award, the Qualcomm Innovation Fellowship, the Intel Rising Star Faculty Award, and the CISCO Faculty Award.

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

CONTENTS

# A   LLM EXPERIMENT DETAILS

## A.1   SIMULATION TASK GENERATION DETAILS

The held-out simulation task list used to compare and evaluate LLM models is "align-rainbow-along-line", "cylinder-in-colorful-container", "splitting-piles", "stack-cylinder-pyramid", "build-pyramid-in-zone", "block-on-"cylinder-on-pallet", "align-cylinder-in-zone", "construct-symmetric-block-wall", "insert-blue-on-red-cylinder", "rainbow-pyramid". We will prompt different LLMs with these task names and run targeted task generation evaluation experiments on code generation three times.

We have also experimented with a longer LLM pipeline with components such as API reviews and common error reviews. In these two stages, we provide the base task definition of the Ravens benchmark and summarize several syntax errors that GPT tends to make when designing and implementing tasks. We empirically observed these to generate more stable codes for challenging tasks, but require more budgets.

## A.2   EMBEDDING ANALYSIS

In Figure 5, we visualize the GPT embedding of the generated task codes using PCA to project to 50 dimensions and then use TSNE for visualization on 2D. We then apply KNN clustering to discover 6 clusters of the generated tasks. As shown in the plot, different colors of clusters represent different groups of tasks such as specific objects or specific types of motions. For instance, the purple color represents mostly the tasks involving rope, and the blue color represents mostly the tasks that involve building some structures using primitive shapes. Inspecting, expanding, and cleaning this task library is also straightforward because of the nature of the code. This embedding space is useful both for retrieving tasks as retrieval augmented generation (RAG) and finding related tasks for joint policy training.

## A.3   TASK STATISTICS

Although we focus on the task-level diversity and generalization of simulation in LLM, in this section, we investigate some task statistics of the 120 generated tasks by LLM. Shown in Figure 9 (a), we observe an interesting balance of colors, assets, actions, and number of instances generated by the LLM models. For example, common colors such as "red, blue, yellow, green" show up in many tasks. Due to our prompt, overall GPT generates multiple objects in the scene. In Figure 13, we also showed a gallery of the generated tasks. We can easily generate and scale more of the simulation tasks if the budget permits. See project website for animated rendering of the tasks.

## A.4   TRAINING DETAILS

In addition to the prompting pipeline, we explored different training and finetuning schemes on LLMs. Different from designing a full prompting techniques to improve GPT4 base model, finetuning focuses on domain-specific tasks such as given the task name, and output its implementation. It has several benefits such as cheaper costs and faster inference speed. We use the 112 tasks as the dataset for finetuning experiments. We use the task names with a short prompt as input tokens and the task code as output tokens for autoregressive training. For closed-source LLM models such as GPT-3 (davinci), GPT-3.5, we use OpenAI's finetuning API for training and inferencing. For Code-LLama experiments (Rozière et al., 2023), we use the open-source Code-LLaMA-Instruct 7B and Code-LLaMA-Instruct 13B models. We use Huggingface transformers for LoRA (Hu et al., 2021) finetuning with quantization and parameter-efficient finetuning for 10 epochs on 2 V-100 GPUs. The training on GPT4 generated dataset with batch size=1 takes around 4 hours to finish 3000 iterations. Generated examples for these open-sourced models can be found in §D.

In evaluating the open-sourced code-llama model, we were not able to get sensible results with just prompting. Specifically, it generates the example code in the prompt for all prompts in the few-shot case or random answers in the zero-shot case. With finetuning, the model writes much more structured code and can achieve similar performance as GPT-3.5. Overall we see improved

code generation with the finetuning models and leave self-bootstrapping and training on more code examples for future work.

### A.5 TASK CREATION LIMITATIONS

There are several common failure modes in the task description and implementations generated by LLM that need to be aware of. These raise attention for inspections before potential downstream applications and also hopefully shed light on future directions. We recommend users to play with to get a sense of the generation process and limitations. Here, we provide detailed descriptions and examples below.

1. **Repeated Tasks:** GPT can generate tasks that have different names (and even different implementations) but represent the same tasks, such as "color-sequenced-block-insertion", "put-block-in-bowl" and "color-coordinated-ball-insertion-in-bowls", which all aim to instruct the robot to put some colored blocks into colored bowls. Similarly, "color-linked-ball-bowl-ordering", "color-coordinated-sphere-and-bowl-match" and "colored-bowl-ball-sorting" are also same tasks.

2. **Mismatched Language Instructions:** GPT can generate mismatched language descriptions for tasks. For instance, for task "color-coordinated-cylinder-stand-in-line", the language instructions try to set goals for each step to put the corresponding colored cylinder on the stand, such as "place the red cylinder on the red stand". However, the task implementation tries to stack as many cylinders as possible onto the same block. This problem is also not captured by the metric used in the benchmark.

3. **Ungrounded Motion Sequence:**. For instance, the task "color-coordinated-bowl-stacking" has the description "put each block in the bowl of the same color and stack the bowls from largest to smallest in the sequence of red, blue, green, and yellow", and yet the action to stack bowls, is neither implemented nor achievable in the benchmark. Similarly in task "ball-in-container" with language instruction "Use the spatula to lift a ball over a wall of boxes and drop it into a container", the action of using a spatula to lift an object is not grounded well. Finally in "push-block-into-container", the motion of pushing a block into some other objects are unlikely to succeed.

4. **Noisy Descriptions:**. There are usually some noises in the task names, task descriptions, and language instructions. For instance "ball-in-bowl-obstacle-course" has "obstacle-course" which is not capturing the task goal. The task descriptions also contain sentences such as "The task requires the robot to pick up four balls of different colors (red, blue, green, yellow) and place each of them into the corresponding colored bowls strategically positioned at different corners of the maze, without knocking over any blocks, demanding careful navigation and color coordination", which is not in its most concise form. Similarly, "rearrange, sort, insert, place" are usually used interchangeably by LLM.

5. **Imbalanced Tasks:**. Due to the bootstrapping process, LLMs has a certain bias towards the majority of the tasks in the task library, especially if the LLM reflection/filtering is turned off. For instance, we usually see "color-coordinated" and pick-place motions in the task generation whereas tasks involving rope and piles are more rarely. For improved success rates, we also do not consider tasks that are overly complicated such as palletizing boxes and tower of haoni. Moreover, the definition of a task is still subjective in robotics. Different tasks such as "arrange,insertion,put" end up using the pick and place motions, and the hope is LLM can help distinguish among different tasks. We limit the benchmark to 100 tasks for a balance of diversity and control of quality.

6. **Task Complexity:** Overall this work only explores top-down pick and place tasks where the demonstration action can be parametrized by pick and place actions. More dexterous tasks in robotics (higher degrees of freedom, more contact-rich motions etc) and reward designs for automatically figuring out how to solve these tasks will be interesting future works.

## B   SIMULATION TRAINING EXPERIMENT

We follow the original CLIPort(Shridhar et al., 2022) setup for training. All simulated experiments are performed using a Universal Robot UR5e equipped with a suction gripper. This setup offers a

Repeated Tasks          Mismatched Languages          Ungrounded Actions

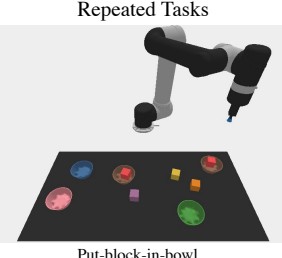
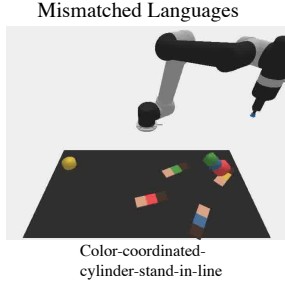
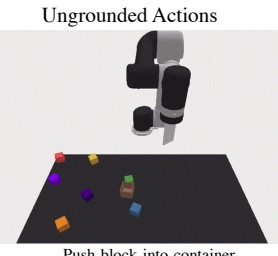

Put-block-in-bowl          Color-coordinated-          Push-block-into-container
                           cylinder-stand-in-line

Figure 10: Visualization of common failure cases.

systematic and replicable environment for evaluation, particularly for benchmarking the capacity to ground semantic concepts like colors and object categories. The input observation is derived from top-down RGB-D images captured via a RealSense camera. We downscale the original image dimensions from $640 \times 480$ to $320 \times 160$ for policy input. To enhance the transition from simulation to reality, we incorporate data augmentation techniques such as color jittering.

Note that our implementation supports multiple samples in a batch (the original version only supports batch size 1) which speeds up training. We train on 4/8 GPUs for 2 days for the multitask setup and train for 10000 iterations for the single-task setup. For larger-scale pretraining experiments, 100 tasks can take up to 5 days with 8 GPUs.

When training language-conditioned multitask policies, the relations among tasks need to be considered in the joint dataset. For instance, the tasks "put-block-in-bowl" and "place-red-in-green" both involve putting colored blocks into colored bowls with similar language instructions. This could potentially lead to conflicting data points that instruct placing the red block into the green bowl or into bowls of other colors. Surprisingly, we find the policies pretrained on all 10 CLIPort tasks only achieve lower than $5\%$ success rate in GPT tasks. Conversely, there could be many tasks that involve the term "pyramid" in instructions, but this could refer to varying layers of the block pyramid. This ambiguity could pose a challenge when training a single policy for multiple tasks. The question of whether training with certain tasks can help generalization can be best addressed by the bias-variance trade-off fundamental in machine learning. Tasks that are semantically closed have a higher likelihood of being beneficial, and vice versa. Therefore, we leverage the task code embedding to pick related tasks for joint training, and find that to be robust in finding tasks. During the evaluation of our GPT-generated tasks, we exclude specific tasks that fall outside of the task generation domains, such as the tower of Hanoi and inserting kittings. In addition to the consideration of compute budgets, these tasks also help us focus more on task-level generalization. The few-shot generalization experiments have been evaluated with 1-shot generalization performance on tasks such as "mixing-piles" and 5-shot generalization performance on "block-insertion".

## C  REAL WORLD EXPERIMENT

For our real-world experiments, we employ an XArm-7 robot equipped with a suction gripper. A bird's-eye-view camera, mounted overhead and oriented downwards, is used to capture RGB-D observations. We leverage workspace boundaries and background subtraction to generate a mask on the depth images. See Figure 11 for a detailed setup. In the real world, we observe the following failure cases: (1) manipulate with the wrong color of objects. (2) imprecise picking and placing. To tackle the challenge of scene-level generalization, we use data augmentation such as color jittering, and also data relabeling where we increase the data multiple times and relabel the language prompt based on the objects on the table. Our labeling tools on pick and place motion do not require a real robot and can efficiently label up to 100 images within 2 minutes. Real-world training takes 50 epochs of the augmented data and usually takes less than 3 hours to complete. The tasks are selected with a focus on long-horizon tasks and task-level generalization, i.e. how well the policy generalizes to new tasks rather than different colors or object instances in a compositional sense. We have experimented with providing both the per-step instruction or high-level goal of the tasks, and did not find much performance difference in the single-task setting. We thus share similar limitations

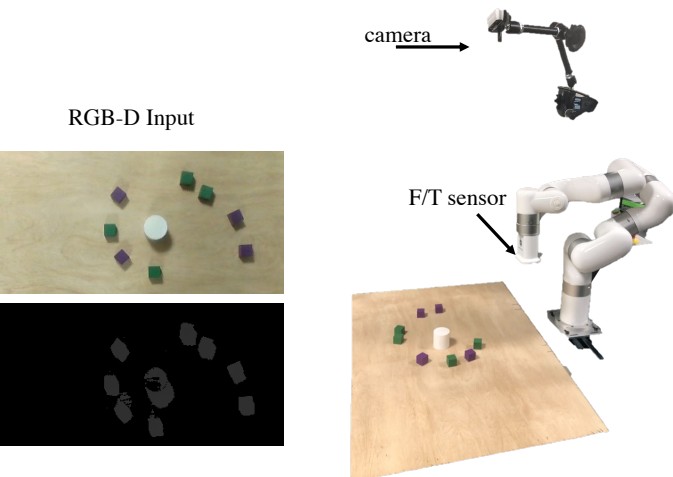

Figure 11: Realworld Setup. We have an overhead camera for observing top-down rgb-d images similar to the simulation setup.

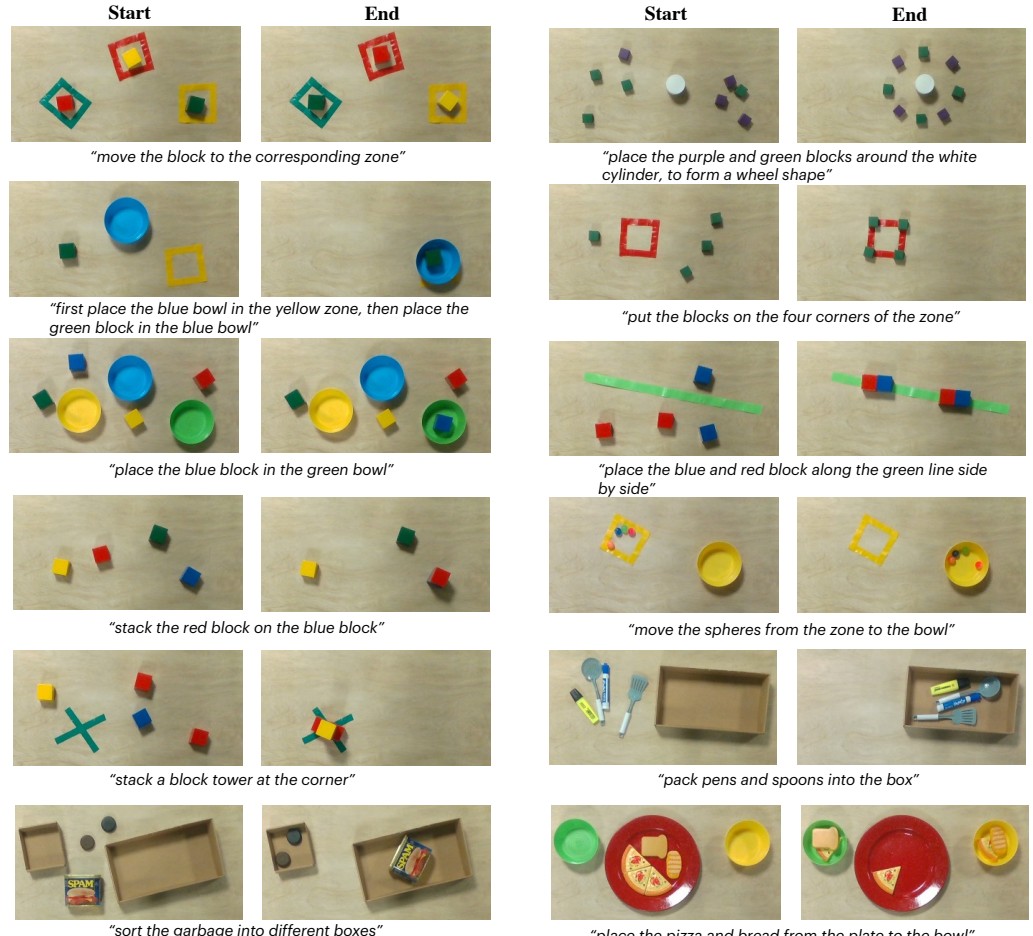

Figure 12: We show the start and goal of the 12 tasks in the real-world experiments. The set involve long-horizon tasks and tasks with very distinct assets from the simulation.

on the policy side as in Shridhar et al. (2022). We note that the main limitation for this choice of pick-and-place task is due to hardware (only suction cup) at the time of experiments.

# D LANGUAGE QUALITATIVE RESULTS

## D.1 TASK DESCRIPTION EXAMPLE

We attach a few generated task description examples from GPT-4.

---

**Task: Build-Car**

- task-name: build-car
- task-description: Construct a simple car structure using blocks and cylinders.
- assets-used: [ block/block.urdf, ball/ball-template.urdf ]

---

**Task: Color-Coordinated-Ball-Insertion-In-Boxes**

- task-name: color-coordinated-ball-insertion-in-boxes
- task-description: Pick up four balls of different colors (red, blue, green, and yellow) and insert them into four separate boxes of matching colors. The boxes are located inside corners of a square structure and the task requires precise insertion, color coordination, and careful navigation around the square structure.
- assets-used: [ ball/ball-template.urdf, box/box-template.urdf, square/square-template.urdf ]

---

**Task: Color-Coordinated-Zone-Arrangement**

- task-name: color-coordinated-zone-arrangement
- task-description: On the tabletop, there are nine blocks of three different colors (three red, three blue, and three green) and three pallets of matching colors (one red, one blue, one green). The task is to pick up each block and place it on the pallet of the same color, arranging the blocks on each pallet in a line. However, there are a few small blocks randomly scattered on the tabletop, which the robot has to navigate around without knocking them over while transporting the blocks to the corresponding pallets. The challenge lies in the precise navigation, placement of the blocks, color matching, and maintaining the balance on the pallets.
- assets-used: [ block/block.urdf, pallet/pallet.urdf, block/small.urdf ]

---

**Task: Place-Blue-On-Line-Ends**

- task-name: place-blue-on-line-ends
- task-description: Pick up each blue box and accurately place it at the end of a green line.
- assets-used: [ line/line-template.urdf, box/box-template.urdf ]

---

**Task: Move-Kit-From-Zone-To-Cylinder**

- task-name: move-kit-from-zone-to-cylinder
- task-description: Place the specific kit from a zone to a cylinder.
- assets-used: [ cylinder/cylinder-template.urdf, zone/zone.urdf, kitting/object-template.urdf ]

---

**Task: Put-Blues-Around-Red**

- task-name: put-blues-around-red
- task-description: Place the blue blocks around the red block.
- assets-used: [ block/block-for-anchors.urdf ]

---

---

**Task: Multicolor-Block-Bridge**

- task-name: multicolor-block-bridge
- task-description: Build a bridge by stacking three red, three blue, and three green blocks on a pallet. Arrange in a sequence from left to right: red, blue, and green. Then, place three cylinders of corresponding colors on top of the stacked blocks, forming a bridge. The cylinders should roll from the top block to the pallet, creating a challenge of precision and control.
- assets-used: [ block/block.urdf, pallet/pallet.urdf, cylinder/cylinder-template.urdf ]

---

## D.2 PROMPT DETAILS

### D.2.1 TASK PROPOSAL PROMPT DETAILS

To reach this level of lengthy code generation, we break the prompt of the agent into several steps to enforce its logical structure: task description generation, API and common mistake summary, few-shot reference code selection, and code generation. The input prompt to GPT-4 task generation stage consists of several components:

(1) available assets that are in the codebase
(2) samples of reference tasks from the task library to act as few-shot examples
(3) past task names to make sure the agent does not provide overlapped tasks.
(4) some examples of bad tasks and the reasons such as not physically feasible.
(5) some additional rules (such as do not use unavailable assets ) and the output format.

This stage has temperature=1 to encourage diversity and the rest components would have temperature=0 to have some robustness. The simplified task description prompt can be found below.

---

**Bottom Up Task Creation Prompt**

You are an AI in robot simulation code and task design. I will provide you some example tasks, code implementation, and some guidelines for how to generate tasks and then you will help me generate a new task. My goal is to design diverse and feasible tasks for tabletop manipulation. I will first ask you to describe the task in natural languages and then will let you write the code for it.

Here are all the assets. Use only these assets in the task and code design.
...

Here are some examples of good tasks. Try to learn from these structures but avoid overlapping with them.
...

Here are some tasks that you have come up with before. Try to learn from these structures but avoid overlapping with these tasks. For instance, 'bowl-ball-placement' and 'sort-balls-in-bowls' are the same task. 'pile-boxes-in-corner' and 'stack-blocks-into-pallet' are similar tasks, 'align-cylinder-in-corner' and 'align-cylinder-corner' are similar.
PAST-TASKNAME-TEMPLATE

Here are some bad example task instances with explanations.
...
reasons: not interesting because it overlaps with the current task 'put-block-in-bowl'.
...

Now please describe the new task in natural languages and explain its novelty and challenges. Format the answer in a python dictionary with keys "task-name" and value type string, "task-description" (one specific sentence) and value type string, and "assets-used" and value type list of strings. Note that
- Do not use assets that are not in the list above.
- Tasks that have more colors and shapes are interesting.
- Be as specific as possible about the number, shape, and color of each asset in the descriptions.
- The task need to obey physics and remain feasible.

---

---

**Top Down Task Creation Prompt**

You are an AI in robot simulation code and task design. I will provide you some example tasks, code implementation, and some guidelines for how to generate tasks and then you will help me generate a new task. **My goal is to design creative and feasible simpler tasks to eventually help solve the task 'TARGET-TASK-NAME'.** I will first ask you to describe the task in natural languages and then will let you write the code for it.

Here are all the assets. Use only these assets in the task and code design.
...

Here are some examples of good tasks. Try to learn from these structures but avoid overlapping with them.
...

Here are some tasks that you have come up with before. Try to learn from these structures but avoid overlapping with these tasks. For instance, 'bowl-ball-placement' and 'sort-balls-in-bowls' are the same task. 'pile-boxes-in-corner' and 'stack-blocks-into-pallet' are similar tasks, 'align-cylinder-in-corner' and 'align-cylinder-corner' are similar.
PAST-TASKNAME-TEMPLATE

Here are some bad example task instances with explanations.
...
reasons: not interesting because it overlaps with the current task 'put-block-in-bowl'.
...

The goal is to solve the task 'TARGET-TASK-NAME' eventually. **Due to its complexity, let's think step-by-step about what simpler task can be useful to achieve this goal. Please describe the new task, which is not 'TARGET-TASK-NAME' but can help training a policy to generalize torwards it, in natural languages in a clear and detailed way. Think step by step how this task can help contribute to the skills that are quired to solve TARGET-TASK-NAME.** Then format the answer in a python dictionary with keys "task-name" and value type string with lower-case and separated by hyphens, "task-description" (one sentence and do not mention urdf paths) and value type string, and "assets-used" and value type list of strings. Note that
- Do not use assets that are not in the list above.
- Tasks that have more colors and shapes are interesting.
- Be as specific as possible about the number, shape, and color of each asset in the descriptions.
- The task need to obey physics and remain feasible.

---

The simplified task code prompt can be found below.

---

**Task Implementation Prompt**

Now I will provide you some reference code and you can write the code for the task "TASK-NAME-TEMPLATE".

""" Code Ignored """

Do not use libraries, functions, and assets that you don't know. For each object, try to describe its color, size, category in the task first before you write the code. You do not need extra helper functions. Comment the code liberally to explain what each piece does and why it's written that way.
Now write the code for the task "TASK-NAME-TEMPLATE" in python code block starting with "'python. Reminder: TASK-STRING-TEMPLATE

---

### D.2.2 TASK MEMORY PROMPT DETAILS

In addition to storing existing generated tasks in the memory, the task reflection stage prompt has the following component:

(1) the generated task description and code

(2) the current tasks in the library

(3) some examples of accepting and rejecting new task

Then LLM is prompted to answer whether to accept this new task and this improvement will also be in the context window in the next round of the agent task design. Note that to improve the robustness of this stage, we prompt GPT three times in parallel to get diverse answers with temperature 0.5, and only accept the task if there is an agreement. The simplified task reflection prompt can be found below.

---

**Task Reflection Prompt**

Do you think your task is sufficiently interesting to be added as a new task for future tasks given that we already have the following task name and descriptions? Moreover, does the simulation code achieve the goal and the language descriptions in the task? Be as rigorous and high-standard as possible.

Reminder: your task: TASK-STRING-TEMPLATE
TASK-CODE-TEMPLATE
current task list: CURRENT-TASK-NAME-TEMPLATE

Reply explain your reasons and then say True or False, formatted in a python dictionary, do not miss commas or add extra spaces. Here are some examples.

Please incorporate these feedbacks when you design new task!

---

The finetuned model usually only requires less tokens and complex prompt as input. Here is an example of the prompt for the finetuned model.

---

**Finetuned Model Task Implementation Prompt**

Write the pybullet simulation task class [TASK_NAME_TEMPLATE]. Provide answers in a python code block starting with "'python

---

### D.3 EXAMPLE TASK LIBRARY CONVERSATIONS

We have observed interesting reference task selection in the task memory to show how the memory can benefit the task creation process.

---

**Task Reference Conversations for "Color-Coordinated-Arch-Construction"**

Question: Please pick 4 task python files that you would like to use as reference. Format them in a python list.

Answer: Sure, based on the task description, the following tasks seem to be the most relevant:

[ color-coordinated-block-tower.py, stack-color-coordinated-blocks.py, construct-colorful-arch.py, color-coordinated-insertion.py ]

---

We also have observed interesting task descriptions and code critics from the task memory. This shows that the critic has looked into the code implementations and reaching an agreement indeed requires the task to be sufficiently high-standard. In this example, the task "color-coordinated-insertion-rearrange" is described as a new task that would rearrange the blocks into a different fixture, but the code and its purpose is exactly the same as the "color-coordinated-insertion". The first critic thinks that the task is repeated with minor differences, while the second critic catches the coding error and refuses to add it to the task library.

---

**Task Reflection Conversations for "Sphere-Insertion"**

Input:

- task-name: color-coordinated-insertion-rearrange
- task-description: There are four colored ell objects (red, blue, green, yellow) inserted in randomly ordered fixtures on the tabletop. The task is to remove each ell object from its current fixture and insert it into a new fixture in a specific color order: from left to right, insert red, blue, green, and finally yellow.
- assets-used: [insertion/ell.urdf, insertion/fixture.urdf]

Critic 0:
Add to task library?: No
Reasons: The task is very similar to "color-ordered-insertion" which also involves inserting colored ell objects into fixtures in a specific order. The only difference is that in the existing task, the ell objects are matched with fixtures of the same color, while in the proposed task, the ell objects are rearranged in fixtures of different colors. However, this difference is minor and does not add significant novelty or complexity to the task. Therefore, it may not be necessary to add the proposed task to the task list.

Critic 1:
Add to task library?: No
Reasons: The task is similar to "color-ordered-insertion" but adds a new challenge of rearranging the ell objects. This task would require more complex manipulation skills and could be interesting for developing advanced robotic skills. However, the simulation code does not seem to ensure the color order from left to right as described in the task. It randomly places the fixtures and ell objects, and the goals are added in the order of the colors array, not according to their positions on the tabletop

---

## D.4   EXAMPLE IMPLEMENTATION ERRORS

We include additional misaligned task and language instruction examples as well as error messages in this section. Specifically, we have collected an "error book" to be used as a reflection on LLM's failures and also include into prompts to improve task generation success rates. Here we show a selected list of concrete examples and our analysis.

1. using assets (urdfs) that do not exist

2. use ambiguous language descriptions as goals. For instance, "place the colored blocks into the matching colored bowls" with one goal and sparse reward as the task instead of adding subgoal "place blue block into blue bowl" and give continuous reward.

3. 'matches' in the goal has wrong dimensions. It should have the same dimensions as number of objects (N) multiplied by the number of goal poses (M). Usually it is N by M.

4. have vector dimension problem such as 'np.random.choice(box_size)' or 'box_size / 2' where 'box_size' is a tuple and not an int

5. make too large an object for stacking or make the task objects invisible for picking.

6. accessing index out of bound 'pallet_pose[2]' for 'pallet_pose'. 'pallet_pose=get_random_pose' returns a tuple (translation, rotation). It does not have 3rd component. Similarly accessing 'container_pose[2]' or 'box_pose[2]' would cause errors as well. Since it's a tuple, try to modify it in-place will also trigger errors.

7. forget to replace str using 'fill_template()' for urdfs with template such as 'cylinder-template.urdf'. 'ball-template.urdf', 'line-template.urf'.

8. use 'self.ee = Spatula()' as a function when doing pushing tasks, which is incorrect. It should be 'self.ee = Spatula'.

9. mistakenly use 'random_pose' for target pose. Design target poses based on task objectives.

10. add only one or fewer language goals which cause language-motion inconsistency. Note that the language goals usually are the same number as the pick and place goals.

Some concrete syntax errors are listed:

```
Syntax Errors

    - "environment.py, line 195", in reset
      self.task.reset(self)
   File "\<string\>, line 38",\\ in reset
TypeError: can only concatenate str (not "list") to str

    - "environment.py, line 195",\\ in reset
   object\textunderscore shape =
       np.random.choice(object\textunderscore shapes)
   in numpy.random.mtrand.RandomState.choice
ValueError: a must be 1-dimensional

 - "task.py, line 325",\\ in get\textunderscore
      random\textunderscore pose
      pos = (pos[0], pos[1], obj\textunderscore size[2] / 2)
IndexError: tuple index out of range

  - "task.py", line 200, \\ in reward
      pts = obj\textunderscore pts[obj\textunderscore id]
IndexError: arrays used as indices must be of integer (or
      boolean) type

  - "task.py", line 206, \\ in reward
        for zone\textunderscore idx, (zone\textunderscore pose,
           zone\textunderscore size) in enumerate(zones):
TypeError: 'NoneType' object is not iterable

  - "task.py", line 252, \\ ball\textunderscore pose =
      self.get\textunderscore random\textunderscore pose(env,
      ball\textunderscore size)
ball\textunderscore pose[0][2] += 0.02
TypeError: 'tuple' object does not support item assignment
```

## D.5 GENSIM FEEDBACKS

We detail the various feedback forms, which have a hierarchical structure, in GenSim in this section. Specifically, once a simulation code is generated by GenSim, it will first be checked on syntax errors by "evaluating" the code piece in python. Then we run through the code to generate demonstrations. During this runtime, we check both the runtime errors (such as missing paths or run out of bound), and if the code can successfully generate demonstrations. Successfully generating the demonstrations requires having the actual robot execution to correctly achieve each subgoal and attain the full reward. This test is unique to robotic simulation, and is critical to filter out bad tasks without any real-world risks. The tasks saved at this stage could already be used for training, but as in other ML pipelines, good training performance depends highly on quality. We note that there are three caveats here: (1) a task code can successfully generate demonstrations but is also mundane or repeated (2) it has misaligned instructions and actual demonstrations (3) it can be perfect but the task is too hard to train a policy. Therefore, to ensure the high-quality tasks, our validation stage has LLM critic that automatically reflects on whether the task is interesting. We have single-task policy training to evaluate if a task is good for training. Finally, we have the human verification which only requires a few seconds on average to look at the generated visualization videos without any technical backgrounds, and can be large-scale deployed with mechanical turks. We detail the human efforts required to scale up simulation tasks in the next section.

## D.6 HUMAN VERIFICATION

Despite the impressive performance that we have seen in the GenSim system, it still has gaps to automatically generate infinite "meaningful and diverse" tasks to train robotic policies in simulation. We note that as in other ML systems, data engineering efforts by humans are also inevitable, and

|         | User 1 | | User 2 | | User 3 | |
|---------|--------|------|--------|------|--------|------|
| Task    | Time   | Pass | Time   | Pass | Time   | Pass |
| Average | 12.83  | 0.8  | 5.11   | 0.63 | 17.4   | 0.41 |

Table 3: Human verification time spent and pass rates on tasks generated. The average human time is around 10 seconds (if each task is checked) and the success rates are above $50\%$.

currently general judicious LLM evaluation is still an open research question. Luckily, in the GenSim pipeline, we have various feedbacks before a human has to come into inspection (for instance, in order to even be considered to be saved as a task, the task needs to be able to successfully generate demonstrations and be considered by other LLM critics as interesting). Moreover, even when a human has to come to check if a task is meaningful and interesting, he/she usually only has to look at the generated demo and give a binary answer, which is almost effortless and can be done very efficiently. Actual code modification and manual human tasks are sometimes useful, but collaborative coding with GenSim in the top-down mode will also be helpful in that case. Overall, we generated 30 tasks to benchmark human filtering, and we measured the amounts of human efforts for three different users with varying experiences in robotic simulation and observed that the average time spent on each task was less than 10 seconds. This usually means looking at the visualization videos and checking if the robot actions match the language instructions, rather than actually checking the code. See Table 3 for more details. We note that the human time spent and the pass rates are likely subjective and depend on each person's familiarity with the contexts of robot simulations. One interesting future direction is to comprehensively study how the quality of the generated tasks might affect the quality of the trained policies. For instance, what levels of degrades can be attributed to misaligned language instructions. For example, GPT-4 model sometimes hallucinates concepts such as "boundary" and "ascending size" without actually implementing them in code. This language annotation problem will not show up in single-task policy training, but it might affect multi-task policy training in some ways.

### D.7 FRAMEWORK SCALABILITY DISCUSSIONS

Although we have shown scaling the simulation task generation with foundation models, we discuss how to improve the efficiency and scalability of such systems in the future. We first note that such a system can support both automatic generation (exploratory) and co-pilot generation (goal-directed) and it reduces the technical backgrounds required for making a new simulation task to train robot policies. We usually can let the program run for an hour or a few hours and then check back on the generated tasks. One issue we have observed is that codebase modifications to support cleaned and documented code structures as well as adding fixes to tolerate some common hallucinations is important to improve the success rates. Two things that we plan to explore in the future are larger-scale retrieval augmented generation (RAG) and advanced sampling in task references. These are important to provide contexts to LLM and guide toward generating more balanced tasks. Finally, we note that such a system can already be deployed with Amazon Turk to scale up more tasks.

### D.8 TASK CODE EXAMPLE

We attach a few generated code examples from LLM (library imports ignored). The generated task visualizations are also attached.

**Task: Color-Colored-Insertion**

```python
class ColorOrderedInsertion(Task):
    """Insert differently-colored ell objects into the matching
        color fixture in a specific order."""

    def __init__(self):
        super().__init__()
        self.max_steps = 20
        self.lang_template = "put the {color} L shape block in
            the L shape hole"
        self.task_completed_desc = "done with insertion."
        self.additional_reset()

    def reset(self, env):
        super().reset(env)

        # Define colors and their order
        colors = ['red', 'blue', 'green', 'yellow']
        color_order = {color: i for i, color in
            enumerate(colors)}

        # Add fixtures.
        fixture_size = (0.12, 0.12, 0.02)
        fixture_urdf = 'insertion/fixture.urdf'
        fixtures = []
        for color in colors:
            fixture_pose = self.get_random_pose(env,
                fixture_size)
            fixture_id = env.add_object(fixture_urdf,
                fixture_pose, color=utils.COLORS[color],
                category='fixed')
            fixtures.append(fixture_id)

        # Add ell objects.
        ell_size = (0.04, 0.04, 0.04)
        ell_urdf = 'insertion/ell.urdf'
        ells = []
        for color in colors:
            ell_pose = self.get_random_pose(env, ell_size)
            ell_id = env.add_object(ell_urdf, ell_pose,
                color=utils.COLORS[color])
            ells.append(ell_id)

        # Goal: each ell is inserted into the matching color
            fixture in the correct order.
        for i, ell in enumerate(ells):
            self.add_goal(objs=[ell], matches=np.ones((1, 1)),
                targ_poses=[p.getBasePositionAndOrientation(
                fixtures[i])], replace=False,
                rotations=True, metric='pose', params=None,
                step_max_reward=1 / len(ells),
                language_goal=self.lang_template.format(
                color=colors[i]))
```

**Task: Four-Corner-Pyramid-Challenge**

```python
class FourCornerPyramidChallenge(Task):
    """Construct a pyramid of blocks in each zone with a
       specific color sequence."""

    def __init__(self):
        super().__init__()
        self.max_steps = 20
        self.lang_template = "build a pyramid of blocks in each
            zone with the sequence red, blue, green, and yellow
            from bottom to top"
        self.task_completed_desc = "done building pyramids."
        self.additional_reset()

    def reset(self, env):
        super().reset(env)

        # Add zones.
        zone_size = (0.12, 0.12, 0)
        zone_urdf = 'zone/zone.urdf'
        zone_poses = []
        for _ in range(4):
            zone_pose = self.get_random_pose(env, zone_size)
            env.add_object(zone_urdf, zone_pose, 'fixed')
            zone_poses.append(zone_pose)

        # Block colors.
        colors = [
            utils.COLORS['red'], utils.COLORS['blue'],
                utils.COLORS['green'], utils.COLORS['yellow']
        ]

        # Add blocks.
        block_size = (0.04, 0.04, 0.04)
        block_urdf = 'block/block.urdf'
        blocks = []
        for i in range(4):
            for _ in range(4):
                block_pose = self.get_random_pose(env,
                    block_size)
                block_id = env.add_object(block_urdf,
                    block_pose, color=colors[i])
                blocks.append(block_id)

        # Associate placement locations for goals.
        place_pos = [(0, -0.05, 0.03), (0, 0, 0.03),
                    (0, 0.05, 0.03), (0, -0.025, 0.08),
                    (0, 0.025, 0.08), (0, 0, 0.13)]
        targs = [(utils.apply(zone_pose, i), zone_pose[1]) for
            i in place_pos for zone_pose in zone_poses]

        # Goal: blocks are stacked in a pyramid in each zone.
        for i in range(4):
            self.add_goal(objs=blocks[i*4:(i+1)*4],
                matches=np.ones((4, 4)),
                targ_poses=targs[i*4:(i+1)*4], replace=False,
                    rotations=True, metric='pose', params=None,
                        step_max_reward=1 / 4,
                        symmetries=[np.pi/2]*4,
                        language_goal=self.lang_template)
```

D.9   TOP-DOWN EVALUATION EXAMPLE

In this section, we showcase different LLM's output for an example task in the top-down evaluation.

Task:cylinder-in-colorful-container (Zero-Shot Prompt Before Code-LLAMA-Instruct Fine-tuning)

```python
class CylinderInColorfulContainer(p.Task):
    def __init__(self, cylinder_radius, cylinder_height,
        container_size, container_color):
        self.cylinder_radius = cylinder_radius
        self.cylinder_height = cylinder_height
        self.container_size = container_size
        self.container_color = container_color

    def create_cylinder(self):
        cylinder_id = p.create_cylinder(self.cylinder_radius,
            self.cylinder_height)
        return cylinder_id

    def create_container(self):
        container_id = p.create_box(self.container_size,
            self.container_color)
        return container_id

    def add_cylinder_to_container(self, cylinder_id,
        container_id):
        p.add_object_to_container(cylinder_id, container_id)

    def run(self):
        cylinder_id = self.create_cylinder()
        container_id = self.create_container()
        self.add_cylinder_to_container(cylinder_id,
            container_id)
        p.step_simulation()
        return cylinder_id, container_id
```

**Task:cylinder-in-colorful-container (Few-Shot Prompt Before Code-LLAMA-Instruct Fine-tuning)**

```python
class CylinderInColorfulContainer(Task):
    def __init__(self):
        super().__init__()
        self.max_steps = 3
        self.lang_template = "put the cylinder in the colorful
            container"
        self.task_completed_desc = "done with insertion."
        self.additional_reset()

    def get_random_pose(self, env, obj_size):
        pose = super().get_random_pose(env, obj_size)
        pos, rot = pose
        rot = utils.eulerXYZ_to_quatXYZW((0, 0, np.pi / 2))
        return pos, rot

    def reset(self, env):
        super().reset(env)

        """Add cylinder."""
        size = (0.1, 0.1, 0.04)
        urdf = 'insertion/cylinder.urdf'
        pose = self.get_random_pose(env, size)
        cylinder_id = env.add_object(urdf, pose)

        """Add colorful container to place cylinder."""
        size = (0.1, 0.1, 0.04)
        urdf = 'insertion/container.urdf'
        targ_pose = self.get_random_pose(env, size)
        env.add_object(urdf, targ_pose, 'fixed')

        self.add_goal(objs=[cylinder_id],
            matches=np.int32([[1]]), targ_poses=[targ_pose],
            replace=False,
                rotations=False, metric='pose', params=None,
                    step_max_reward=1, symmetries=[2 * np.pi],
                language_goal=self.lang_template)
```

**Task: cylinder-in-colorful-container (After Code-LLAMA-Instruct Finetuning)**

```python
import numpy as np
from cliport.tasks import Primitive
from cliport.tasks.task import Task
from cliport.utils import utils

class CylinderInColorfulContainer(Task):
    """Pick up each cylinder and place it into the container of
        the same color, in a specific color sequence."""

    def __init__(self):
        super().__init__()
        self.max_steps = 20
        self.lang_template = "place the {color} cylinder in the
            {color} container"
        self.task_completed_desc = "done placing cylinders in
            containers."
        self.additional_reset()

    def reset(self, env):
        super().reset(env)

        # Define the color sequence
        color_sequence = ['red', 'blue', 'green', 'yellow']

        # Add containers.
        container_size = (0.12, 0.12, 0.12)
        container_urdf = 'container/container-template.urdf'
        container_poses = []
        for color in color_sequence:
            container_pose = self.get_random_pose(env,
                container_size)
            env.add_object(container_urdf, container_pose,
                'fixed', color=color)
            container_poses.append(container_pose)

        # Add cylinders.
        cylinder_size = (0.04, 0.04, 0.04)
        cylinder_urdf = 'cylinder/cylinder-template.urdf'
        cylinders = []
        for color in color_sequence:
            cylinder_pose = self.get_random_pose(env,
                cylinder_size)
            cylinder_id = env.add_object(cylinder_urdf,
                cylinder_pose, color=color)
            cylinders.append(cylinder_id)

        # Goal: each cylinder is in the container of the same
            color, in the specified color sequence.
        for i in range(len(cylinders)):
            self.add_goal(objs=[cylinders[i]],
                matches=np.ones((1, 1)),
                targ_poses=[container_poses[i]], replace=False,
                    rotations=True, metric='pose', params=None,
                        step_max_reward=1/len(cylinders),
language_goal=self.lang_template.format(color=color_sequence[i]))
```

**Task: cylinder-in-colorful-container (GPT 3 (davinci) Finetuned)**

```python
import numpy as np
import os
import pybullet as p
import random
from cliport.tasks import primitives
from cliport.tasks.grippers import Spatula
from cliport.tasks.task import Task
from cliport.utils import utils
import numpy as np
from cliport.tasks import primitives
from cliport.tasks.grippers import Spatula
from cliport.tasks.task import Task
from cliport.utils import utils

class CylinderInColorfulContainer(Task):
    """Put a cylinder into a container of a specific color"""

    def __init__(self):
        super().__init__()
        self.max_steps = 10
        self.lang_template = "put the cylinder into the {color}
            container"
        self.task_completed_desc = "done putting cylinder in
            container."
        self.additional_reset()

    def reset(self, env):
        super().reset(env)

        # Add cylinder.
        cylinder_size = (0.02, 0.02, 0.05)
        cylinder_urdf = 'cylinder/cylinder.urdf'
        cylinder_pose = self.get_random_pose(env, cylinder_size)
        cylinder_id = env.add_object(cylinder_urdf,
            cylinder_pose)

        # Add containers.
        container_size = (0.1, 0.1, 0.1)
        container_urdf = 'cylinder/container.urdf'

        # Colors of containers.
        # IMPORTANT: RETRIEVE THE ACTUAL COLOR VALUES
        container_colors = [utils.COLORS[c] for c in
            utils.COLORS if c != 'red']

        container_poses = []
        container_ids = []
        for i in range(3):
            container_pose = self.get_random_pose(env,
                container_size)
            container_id = env.add_object(container_urdf,
                container_pose, color=container_colors[i])
            container_poses.append(container_pose)
            container_ids.append(container_id)

        # Goal: cylinder is in the red container.
        self.add_goal(objs=[cylinder_id], matches=np.ones((1,
            3)), targ_poses=container_poses, replace=True,
                rotations=False, metric='pose', params=None,
                    step_max_reward=1)
        self.lang_goals.append(self.lang_template.format(color="red"))
```

**Task: cylinder-in-colorful-container (GPT3.5 Prompting)**

```python
import numpy as np
from cliport.tasks.task import Task
from cliport.utils import utils

class CylinderInColorfulContainer(Task):
    def __init__(self):
        super().__init__()
        self.max_steps = 10
        self.lang_template = "Place the {color} cylinder in the
            {color} container"
        self.task_completed_desc = "Cylinders placed in
            containers."
        self.additional_reset()

    def reset(self, env):
        super().reset(env)

        # Define colors and corresponding containers
        colors = ['red', 'blue', 'green', 'yellow']
        containers = ['red', 'blue', 'green', 'yellow']

        # Shuffle the colors and containers to avoid always
            picking the same combination
        np.random.shuffle(colors)
        np.random.shuffle(containers)

        # Add containers
        container_size = (0.1, 0.1, 0.1)
        container_template = 'container/container-template.urdf'
        container_poses = []
        for color in containers:
            container_pose = self.get_random_pose(env,
                container_size)
            container_urdf =
                container_template.replace('COLOR', color)
            env.add_object(container_urdf, container_pose,
                'fixed')
            container_poses.append(container_pose)

        # Add cylinders
        cylinder_size = (0.05, 0.05, 0.1)
        cylinder_template = 'cylinder/cylinder-template.urdf'
        cylinders = []
        for i, color in enumerate(colors):
            cylinder_pose = self.get_random_pose(env,
                cylinder_size)
            cylinder_urdf = cylinder_template.replace('COLOR',
                color)
            cylinder_id = env.add_object(cylinder_urdf,
                cylinder_pose)
            cylinders.append(cylinder_id)

        # Add goals
        for i, color in enumerate(colors):
            language_goal =
                self.lang_template.format(color=color)
            self.add_goal(objs=[cylinders[i]],
                matches=np.ones((1, 1)),
                targ_poses=[container_poses[i]], replace=False,
                        rotations=True, metric='pose',
                            params=None, step_max_reward=1,
                            language_goal=language_goal)
```

**Task: cylinder-in-colorful-container (GPT3.5 Finetuned)**

```python
import numpy as np
import os
import pybullet as p
import random
from cliport.tasks import primitives
from cliport.tasks.grippers import Spatula
from cliport.tasks.task import Task
from cliport.utils import utils

class CylinderInColorfulContainer(Task):
    """Pick up a cylinder of a specific color and place it into
        a container of the same color."""

    def __init__(self):
        super().__init__()
        self.max_steps = 20
        self.lang_template = "put the {color} cylinder in the
            {color} container"
        self.task_completed_desc = "done placing cylinders in
            containers."
        self.additional_reset()

    def reset(self, env):
        super().reset(env)

        # Define colors and their corresponding containers and
            cylinders.
        colors = ['red', 'blue', 'green', 'yellow']
        containers = []
        cylinders = []
        for color in colors:
            # Add container.
            container_size = (0.12, 0.12, 0.12)
            container_pose = self.get_random_pose(env,
                container_size)
            container_id =
                env.add_object('container/container-template.urdf',
                container_pose, color=utils.COLORS[color])
            containers.append(container_id)

            # Add cylinder.
            cylinder_size = (0.04, 0.04, 0.04)
            cylinder_pose = self.get_random_pose(env,
                cylinder_size)
            cylinder_id =
                env.add_object('cylinder/cylinder-template.urdf',
                cylinder_pose, color=utils.COLORS[color])
            cylinders.append(cylinder_id)

        # Goal: each cylinder is in a container of the same
            color.
        for i in range(len(colors)):
            self.add_goal(objs=[cylinders[i]],
                matches=np.ones((1, 1)), targ_poses
                =[p.getBasePositionAndOrientation(
            containers[i])], replace=False,
                    rotations=True, metric='pose', params=None,
                    step_max_reward=1/len(colors),
            language_goal=self.lang_template.format(color=colors[i]))
```

**Task: cylinder-in-colorful-container (GPT 4)**

```python
import numpy as np
import os
import pybullet as p
import random
from cliport.tasks import primitives
from cliport.tasks.grippers import Spatula
from cliport.tasks.task import Task
from cliport.utils import utils
import numpy as np
from cliport.tasks.task import Task
from cliport.utils import utils

class CylinderInColorfulContainer(Task):
    """Pick up four cylinders of distinct colors (red, blue,
        green, yellow) and place each of them into a container
        of the same color."""

    def __init__(self):
        super().__init__()
        self.max_steps = 4
        self.lang_template = "put the {} cylinder in the {}
            container"
        self.task_completed_desc = "done placing cylinders in
            containers."
        self.additional_reset()

    def reset(self, env):
        super().reset(env)

        # Define colors
        colors = ['red', 'blue', 'green', 'yellow']

        # Add containers.
        container_size = (0.12, 0.12, 0.12)
        container_template = 'container/container-template.urdf'
        container_poses = []
        for color in colors:
            container_pose = self.get_random_pose(env,
                container_size)
            env.add_object(container_template, container_pose,
                category='fixed', color=utils.COLORS[color])
            container_poses.append(container_pose)

        # Add cylinders.
        cylinder_size = (0.04, 0.04, 0.12)
        cylinder_template = 'cylinder/cylinder-template.urdf'
        cylinders = []
        for color in colors:
            cylinder_pose = self.get_random_pose(env,
                cylinder_size)
            cylinder_id = env.add_object(cylinder_template,
                cylinder_pose, color=utils.COLORS[color])
            cylinders.append(cylinder_id)

    # Goal: each cylinder is in a container of the same color.
        for i in range(len(cylinders)):
            self.add_goal(objs=[cylinders[i]],
                matches=np.ones((1, 1)),
                targ_poses=[container_poses[i]], replace=False,
                        rotations=True, metric='pose',
                            params=None,
                            step_max_reward=1/len(cylinders))
            self.lang_goals.append(self.lang_template.format(
            colors[i], colors[i]))
```
33

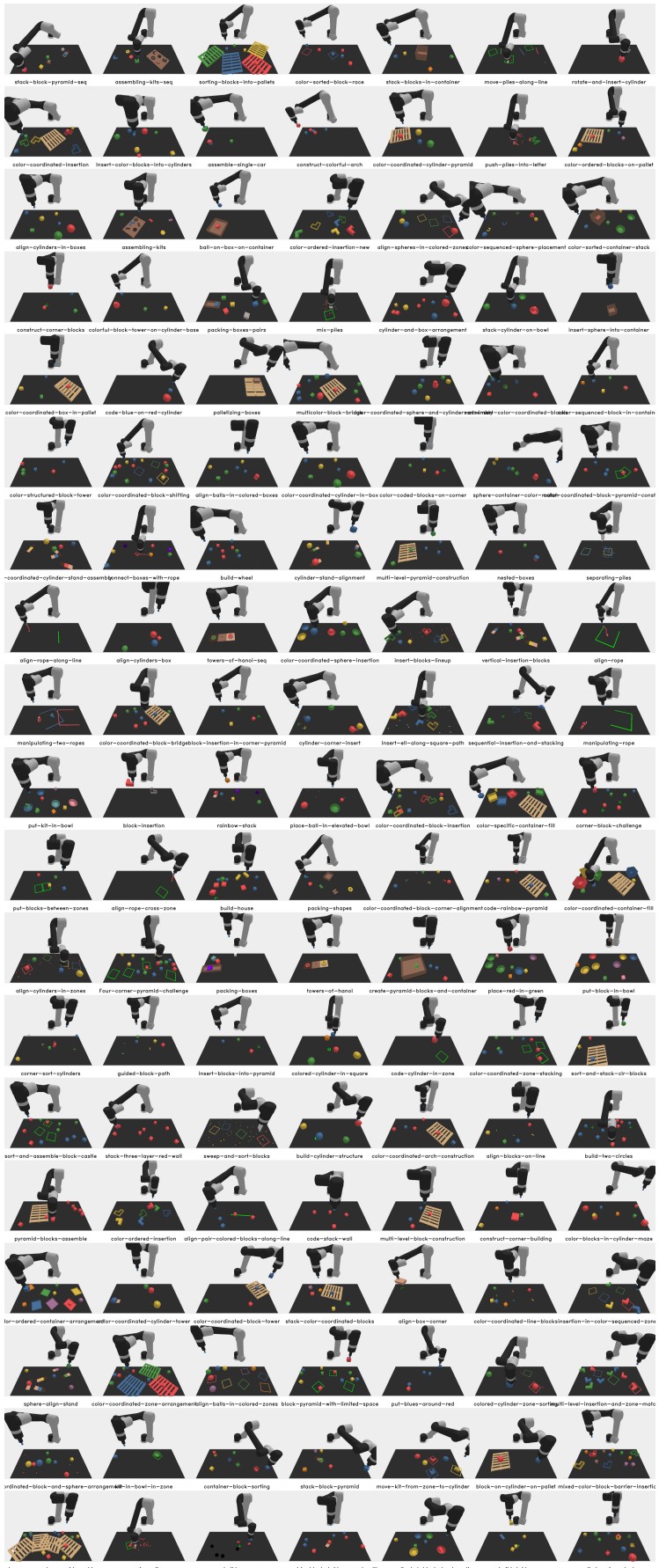

Figure 13: Gallery of generated tasks by GPT-4.

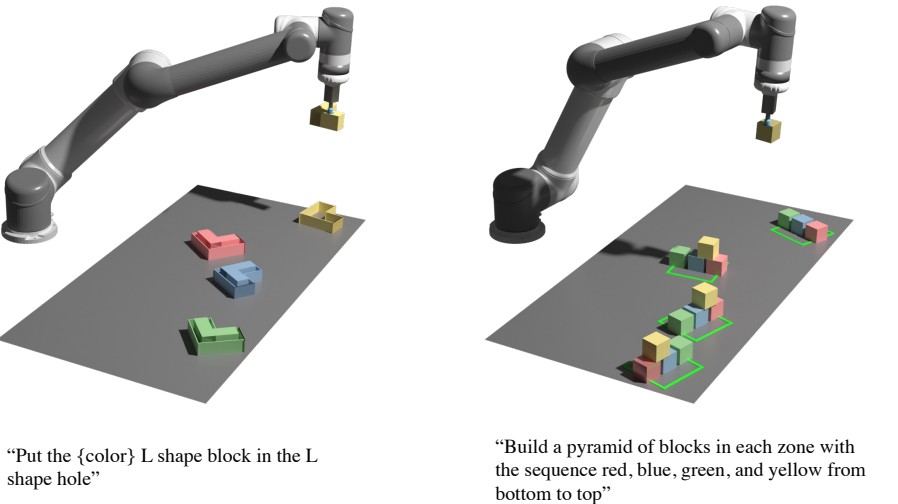

"Put the {color} L shape block in the L shape hole"

"Build a pyramid of blocks in each zone with the sequence red, blue, green, and yellow from bottom to top"

Figure 14: The simulation task visualizations for "color-ordered-insertion" and "four-corner-pyramid-challenge" correspond to the implementations below.

