# OpenReview forum: "GenSim: Generating Robotic Simulation Tasks via Large Language Models"
_ICLR.cc/2024/Conference — ICLR 2024 spotlight_

### Official Review · Reviewer_LF4g · 2023-10-27

**Soundness:** 3 good
**Presentation:** 3 good
**Contribution:** 3 good
**Rating:** 8
**Confidence:** 4

**Summary:**

In summary, this study introduces GenSim, a scalable framework harnessing large language models (LLMs) to enhance robotic policy simulation tasks.

The primary goal is to distill LLM capabilities into practical, low-level policies for robots. We delved into LLM finetuning and prompting methods, exploring both goal-directed and exploratory approaches to generate novel simulation task codes.

The study focused on table-top pick-and-place tasks, and addressing challenges in generating intricate robotic tasks remains an area for future exploration. Moreover, future research could explore algorithms for training multitask policies capable of efficiently adapting to these larger-scale generated task benchmarks, thereby paving the way for more sophisticated and realistic robotic simulations.

**Strengths:**

The authors successfully leverage Large Language Models (LLMs) to tackle the issue of limited task-level diversity in robotic simulation. By tapping into the reasoning and coding capabilities of LLMs, the proposed GenSim framework autonomously designs and validates task asset arrangements and progressions, leading to the creation of over 100 tasks.

Secondly, the paper presents a meticulous evaluation of the generated tasks using tailored metrics, ensuring the tasks' quality and achievability. The authors go further to compare different LLM models, including GPT4 and Code-Llama, and demonstrate that prompting and finetuning techniques based on the task library significantly enhance the LLMs' ability to generate tasks of higher quality.

Additionally, the paper showcases the practical impact of the generated tasks on language-conditioned visuomotor policies. The rigorous evaluation and comparison of different LLM models, and the tangible improvements demonstrated in the generalization capabilities of language-conditioned visuomotor policies.

**Weaknesses:**

Although the paper discusses sim-to-real transfer briefly, there is a lack of in-depth evaluation regarding the effectiveness of policies trained on LLM-generated tasks in real-world robotic applications. Generating over 100 tasks is a positive step, but the computational resources, time, and effort required for this scale of task generation are crucial aspects, especially when considering real-world applications. Without addressing these scalability issues, the framework's practical utility might be limited.

The paper acknowledges the imperfections in the evaluation metrics used for code generation, including misaligned language descriptions. However, it does not delve deeply into the potential implications of these imperfections on the quality of the generated tasks. A more comprehensive analysis of how these imperfections affect task quality and subsequent policy training is essential.

Compared with the vision modules, the language planning modules are much more sophisticated. The potential of this model is not fully unlocked.

**Questions:**

Could you provide more details on the types of syntax errors and lack of grounding observed in the generated code? Understanding these specific issues would be valuable for refining the framework. Additionally, how do these errors impact the subsequent policy training and task performance?

Considering the limitations and challenges identified, what are the key areas of future research you are planning to explore? Are there specific aspects of GenSim that you aim to enhance or refine based on the feedback and insights gathered from this study?

**Details Of Ethics Concerns:**

No ethics concerns

---

> ### Author Response · Authors · 2023-11-19
> **Thank you for your constructive feedback!**
>
> Thank you for finding the topic of our work is rigorous and practical impacts, and appreciating our contribution. We have revised the manuscript and will address your questions below.
>
> > Although the paper discusses sim-to-real transfer briefly, there is a lack of in-depth evaluation regarding the effectiveness of policies trained on LLM-generated tasks in real-world robotic applications. Without addressing these scalability issues, the framework's practical utility might be limited.
>
> Thanks for asking this and we see our work as a first step toward the direction of scalability in simulation and deployment in the real world. We have a full real-world experiment section to discuss how the pre-trained models in simulation might improve the real-world performance in both seen and unseen tasks. We have also added Appendix Section D.7  to discuss specifically this scalability issue. Moreover, we believe that the costs and performance of generating new tasks with foundation models will go down as LLM evolves.
>
> > The paper acknowledges the imperfections in the evaluation metrics used for code generation, including misaligned language descriptions. However, it does not delve deeply into the potential implications of these imperfections on the quality of the generated tasks. A more comprehensive analysis of how these imperfections affect task quality and subsequent policy training is essential.
>
> Thanks for asking this. For instance, misaligned language instructions and tasks can potentially lead to a good single-task policy but potentially can hurt when we conduct joint multitask policies (as it can become a data point with wrong labels). However, we have also observed some robustness in experiments. For instance, when we scale tasks pretraining from 5 to 70 tasks, we did not apply particular per-task noise filtering and the multitask policy seems to generalize well in our evaluation. Finally, we have added some discussions in Appendix Section D.5 and Section D.6 on concrete errors and limitations.
>
>
> >  Could you provide more details on the types of syntax errors and lack of grounding observed in the generated code? Understanding these specific issues would be valuable for refining the framework. Additionally, how do these errors impact the subsequent policy training and task performance?
>
> There are low-level mistakes and high-level mistakes. The low mistakes can be simple syntax errors such as “TypeError: can only concatenate str (not "list") to str'' and “IndexError: arrays used as indices must be of integer (or boolean) type” or failures to set the right goal for task completion. The high-level mistakes can be language misalignment and hallucination. Luckily, we have a full automatic mechanism of task filtering such that these tasks will likely be rejected during the task generation process, and therefore will not impact policy training. We have added concrete error messages (Figure 3) and provided more summaries as well as discussions on how LLM errors propagate to policy training in Appendix D.4 and D.5.
>
> > Considering the limitations and challenges identified, what are the key areas of future research you are planning to explore? Are there specific aspects of GenSim that you aim to enhance or refine based on the feedback and insights gathered from this study?
>
> Thanks for asking this. We have mentioned some limitations and future works in both the main paper and the Appendix. There are three stages in this work, generating the simulation tasks, solving these simulation tasks, and distilling them into a policy. We believe that all of these three problems are in their infancy, and are actively exploring these directions. Automatically generating and scaling these tasks require careful engineering and insights into the LLM pipeline, which has lots of room for improvement. For instance, how to balance between diversity of the tasks and the success rates of generating these tasks. Solving these top-down pick and place tasks is not difficult, but extending to more general dexterous 6 dof tasks are also very challenging problems. Finally, once we have many tasks, how to efficiently distill them into a general policy is also a research question.
>
> Thank you very much again for your constructive feedback. Please do not hesitate to let us know if you have any other questions and/or comments.

---

> > ### Comment · Reviewer_LF4g · 2023-11-22
> >
> > Thanks for your response.
> >
> > Thanks for your efforts in this area.

---

> ### Author Response · Authors · 2023-11-22
> **Thank you for your constructive feedback!**
>
> Dear reviewer, thanks so much for the response and recognizing our efforts. If possible, we wonder if you would kindly consider raising the scores. Thanks tremendously again for the helpful suggestions for improving the paper and happy Thanksgiving!

---

### Official Review · Reviewer_hwME · 2023-10-28

**Soundness:** 4 excellent
**Presentation:** 4 excellent
**Contribution:** 3 good
**Rating:** 8
**Confidence:** 3

**Summary:**

The paper proposes a framework for generating new tasks given a fixed number of existing tasks by leveraging the code generation capability of LLMs for robotics simulation environments. It further uses the generated over 100 tasks for training multi-task policies and evaluates them in both simulation and real world. Stronger generalization is observed by training across more diverse tasks. The idea is novel and profound for breaking the hardness of creating new robot learning tasks in simulation.

**Strengths:**

The task-level diversity is indeed a hard problem due to the required human efforts and this paper proposes a promising approach for solving it, by a novel usage of LLMs. It verifies the effectiveness of this approach and is inspiring.

The strong generalization of the multi-task policies even with zero-shot transfer on new tasks is impressive.

The paper is well-written.

The experiments are thorough for verifying the generalization capability of the policies.

**Weaknesses:**

Most of the current tasks are within the domain of top-down pick and place, and the generalization of policies within such a domain is relatively easy. It could be more impressive to see more dexterous manipulation tasks and policy generation over those.

For the failure cases of task generation, perhaps providing more examples will draw a better boundary on the limitation of the current approach, or for each model.

**Questions:**

In Sec. 3.1, for exploration task generation, which LLM is used? Is it fine-tuned or GPT models without fine-tuning? Also for the generation of 120 tasks, are the newly generated tasks used for fine-tuning as well or just as prompt examples? Please provide more explanations.

For Fig. 6, it seems GPT-4 without fine-tuning outperforms all other fine-tuned models, so is it true that using fine-tuned smaller models is just for consideration of inference cost?

As mentioned in Sec. 1 the task library is initialized with only 10 tasks, for fine-tuning the LLMs to gain context of robotic simulation tasks, is the overfitting to these 10 tasks a problem? If so, how is it handled?

---

> ### Author Response · Authors · 2023-11-19
> **Thank you for your constructive feedback!**
>
> Thank you for finding the topic of our work inspiring and impressive, and appreciating our contribution. We have revised the manuscript and will address your questions below.
>
>
> >  Most of the current tasks are within the domain of top-down pick and place, and the generalization of policies within such a domain is relatively easy. It could be more impressive to see more dexterous manipulation tasks and policy generation over those.
>
> While the domain of top-down pick-and-place, especially short-horizon tasks, are *seemingly* not challenging to most roboticists, we note that the  **task-level policy generalization** explored in GenSim, such as long-horizon tasks and compositional tasks, remains under-explored and non-trivial. For example, "build-a-car" is a nontrivial task that requires semantic understanding of natural language instructions and more than 10 steps of precise pick-and-place and stacking. We are actively experimenting with extending our pipeline to handle more challenging tasks such as dexterous, 6-dof, contact-rich tasks.
>
>
> > For the failure cases of task generation, perhaps providing more examples will draw a better boundary on the limitation of the current approach, or for each model.
>
> Thanks for asking this.  Overall, we acknowledge that even with the strongest model GPT-4 in the current approach, there are still gaps in achieving generation of infinite *high-quality* tasks without any human efforts, due to **the bottleneck of grounding and the hallucination** problems in LLMs at its current state. We have added detailed error messages to Appendix D.4. The GPT-4 model sometimes hallucinates concepts such as “boundary” and “ascending size” without implementing these in code. For weaker models such as Code-LLama and GPT-3.5, the gap from GPT-4 is in generating creative and complex tasks, and also in correctly implementing them in code.
>
>
> > In Sec. 3.1, for exploration task generation, which LLM is used? Is it fine-tuned or GPT models without fine-tuning? Also for the generation of 120 tasks, are the newly generated tasks used for fine-tuning as well or just as prompt examples? Please provide more explanations.
>
> **GPT-4 without finetuning** is used in the exploration task generation to generate the task library. This is because the base GPT-4 model is the current strongest LLM model (See Figure 6 right) and has the desired creativity and reasoning ability to generate new tasks in exploratory task generation stage. In addition to being retrieved in the prompts for GPT-4, these 120 tasks are also used for finetuning the open-source model such as Code LLama and closed-source model such as GPT 3.5. However, as shown in Figure 6 right, even fine-tuning the GPT3.5 or codellama with 120 tasks, the success rate cannot outperform GPT4. Therefore, we choose GPT-4 as the base model to generate gensim tasks.
>
> > For Fig. 6, it seems GPT-4 without fine-tuning outperforms all other fine-tuned models, so is it true that using fine-tuned smaller models is just for consideration of inference cost?
>
> Lower costs, because of fewer prompt tokens and cheaper models, and better efficiency, because of more structured and domain-specific output, are important motivations for finetuning LLMs. Moreover, we show that scaling up simulation tasks can also work well with open-source LLM models, where we have access to the model itself and potentially can be used for many other purposes. This finetuning process also enables self-instruct improvements as the task library is expanding.
>
> > As mentioned in Sec. 1 the task library is initialized with only 10 tasks, for fine-tuning the LLMs to gain context of robotic simulation tasks, is the overfitting to these 10 tasks a problem? If so, how is it handled?
>
> In finetuning experiments, we used **all the 100 tasks** from the generated task library from GPT-4 to finetune LLMs. Indeed, during the LoRA finetuning process, we have observed some overfitting problems. We expect further scaling up training tasks to conduct self-improvement will improve this issue.
>
> Thank you very much again for your constructive feedback. Please do not hesitate to let us know if you have any other questions and/or comments.

---

> > ### Comment · Reviewer_hwME · 2023-11-22
> >
> > Thanks for the response.
> >
> > This is an essential contribution of scaling-up experiments for policy generalization in table-top manipulation.

---

> ### Author Response · Authors · 2023-11-22
> **Thank you for your constructive feedback!**
>
> Dear reviewer, thanks so much for the response and recognizing our efforts. If possible, we wonder if you would kindly consider raising the scores. Thanks tremendously again for the helpful suggestions for improving the paper and happy Thanksgiving!

---

### Official Review · Reviewer_1vRz · 2023-10-31

**Soundness:** 3 good
**Presentation:** 3 good
**Contribution:** 3 good
**Rating:** 8
**Confidence:** 3

**Summary:**

This paper addresses the problem of automatically generating robotic tasks in synthetic simulation environments. A method relying on LLMs is presented, where a LLM is used to generate the code that describes a task (assets selection, initial configuration of the assets, and language/robot motion instructions). A task library is initialized using 10 examples written by a human, and the LLM is asked to generate new tasks given carefully designed prompts. The generated tasks are added to the task library if they pass validation tests (e.g. the syntax is correct, the success rate of a policy trained on this task is high enough, it passes some human inspection etc), or the feedback is explicitly given to the LLM as additional instructions in a few-shot prompting scheme. 120 tasks are generated and policies are pre-trained on these tasks. The experimental evaluation shows that pre-training on a large number of tasks increases zero-shot generalization performance on new tasks (held-out test set also generated) in both simulation and real environments after fine-tuning.

**Strengths:**

S1. The problem addressed in this paper is important. Multi-task robot policies strongly rely on diverse training data (in terms of environments and task) but access to such training data is limited by real collection efforts, or having to manually invent and design new tasks. Automatically generating such training data could have a significant impact.

S2. The overall method is technically sound. The appendix provides many details. The details of the prompts which are provided to the LLM is appreciated. Their design design is key as demonstrated in Fig 6 left). However, additional intuition why the two-stage prompt chain is better than the single-prompt would be appreciated.

S3. The evaluation of Table 1 is convincing. In this experiment, multiple models pre-trained on synthetic data are fine-tuned using few real demonstrations. The models pre-trained on 70 generated tasks performs significantly better than a model pre-trained on 50 generated tasks and on the 10 original tasks from CLipport. This experiment demonstrates the practical applications of the proposed approach.

**Weaknesses:**

W1. Figure 3 mentions that human inspection of a generated task is required. Was feedback collected for each of the generated tasks ? How many tasks were manually annotated in total. In addition, what is the granularity of feedback provided by the human ?

W2. The experiments presented in Table 1 use a different set of tasks than the real tasks used to evaluate CLIPort. In particular, the tasks do not involve sweeping, or folding of objects. The variety of asset is also smaller than what is shown in CLIPort where scissors or chess pieces are manipulated. A similar observation is made regarding the synthetic experiments, e.g. the held-out simulation tasks used for evaluation mostly use blocks and omit more complex objects like ropes. Additional explanations on how these tasks were selected are needed to convince the readers that the method indeed generalizes to a wide variety of tasks. Experiments presented in Fig 6 left) evaluate models on the same tasks as CLIPort but it is not clear if the models were pre-trained on all generated tasks in this experiment (see W3.)

W3. The “few-shot policy generalization to related tasks” experiment needs additional description to understand the meaning of “single-original, single+2, single+4 etc”. Were the models pre-trained on original clipport tasks, original clipport tasks + 2 generated tasks etc ? Or are the models pre-trained on all CLIPort tasks + all the ones that were generated automatically ? It is important to clarify this experiment as the current version lets the reader think that only a small subset of of the generated tasks were used to pre-train the policy, and the question of how these were selected arises.

W4. Page 6 mentions “When only pretraining on the 10 tasks in clipport, the policy does not generalize well on the GPT4 tasks.” In fig 7 right) the model pre trained on tasks generated with LLMs is better than the one trained on CLipport tasks, but the difference is 15-20% of relative improvements, the claim should therefore be mitigated. It is also suggested to justify how the set of testing tasks (given in Appendix A.1) was selected. See also W2.

 W5. Details of the metrics used in Fig 6 would be appreciated. In particular, how is “code reasoning capability” measured in “runtime-verified”, and is the “task completed” an average over several environment resets ?

**Questions:**

The paper addresses an important problem and the method is technically sound. The experiments demonstrate that policies pre-trained on the automatically generated tasks can generalize to novel tasks in both synthetic and real environments. I am however concerned about the diversity of the tasks which the method is evaluated on. In particular, the evaluation tasks seem to be less varied compared to the ones used in CLIPort (W2). I am also questioning the “automatic” nature of the method as human intervention is required (W1). I will gladly update my review if the authors can address these concerns and respond to my questions (see weaknesses).

---

> ### Author Response · Authors · 2023-11-19
> **Title: Thank you for your constructive feedback! (1)**
>
> Thank you for finding the topic of our work is an important problem and the method is technically sound, and appreciating our contribution. We have revised the manuscript and will address your questions below.
>
>
> > Additional intuition why the two-stage prompt chain is better than the single-prompt would be appreciated.
>
> Creating simulation task descriptions and implementing the code for the tasks require different language capabilities. For creating the task, we would like to generate tasks as *creative and interesting* as possible (with high temperature for decoding) while being physically feasible, so it tests the language reasoning capabilities of LLM. For implementing the task code in simulation, it requires *rigorous coding execution* (with zero temperature for decoding) and needs to be thoughtful in the details of syntax and goal generation. Therefore a two-stage prompt chain with two subtasks is favored. We also have an ablation study on the task performance generated by single-shot and two-stage prompts in Figure 6 to show that a two-stage prompt chain with few-shot examples and task library can effectively improve the code generation success rates.
>
>
> >W1. Figure 3 mentions that human inspection of a generated task is required. Was feedback collected for each of the generated tasks ? How many tasks were manually annotated in total. In addition, what is the granularity of feedback provided by the human ?
>
> We note that human inspection is *not required* for each task and the effort of annotation only involves watching rendered videos (without the need for checking code).  First, around 90% of erroneous tasks can be filtered out via automatic metrics (e.g., execution errors). Second, for the rest of the tasks, humans only need to check whether the simulated tasks are correct or not by checking whether the rendered videos correspond to the natural language description of the tasks.  In our experiments, LLMs generate around *200* tasks in total, and humans (i.e. the authors) checked and filtered around 150 tasks and kept 100 tasks.  Our new experiment below also shows that the human filtering rate is around 50%.
>
> During the response period, we assess human effort (time and pass rates) quantitatively by re-generating 30 tasks for human feedback. In Table 3 (attached below, also in Appendix Section D.6), we find that the required human efforts are around 10 seconds (for CS students) on average for each task. Note again that this is 10% auto-filtered tasks and the amounts of total inspectations are much smaller than all generated tasks. For future work, we are also exploring whether non-CS humans can check the correctness so that we can scale up the process of collecting human feedback.
>
> | Task       | User 1 Time | User 1 Pass | User 2 Time |User 2  Pass | User 3 Time |User 3 Pass |
> |------------|------------|------------|------------|------------|------------|------------|
> | Average    | 12.83 | 0.8 | 5.11  | 0.63 | 17.4 | 0.41 |
>
> You can also get a sense of how much effort is needed for verification by trying out our Gradio demo (https://huggingface.co/spaces/Gen-Sim/Gen-Sim).

---

> ### Author Response · Authors · 2023-11-19
> **Thank you for your constructive feedback! (2)**
>
> > W2. The experiments presented in Table 1 use a different set of tasks than the real tasks used to evaluate CLIPort. In particular, the tasks do not involve sweeping, or folding of objects. The variety of assets is also smaller than what is shown in CLIPort. A similar observation is made regarding the synthetic experiments, e.g. the held-out simulation tasks used for evaluation mostly use blocks. Additional explanations on how these tasks were selected are needed.
>
>
>  We have conducted additional experiments to verify that our framework can also extended to more varied tasks and assets as suggested. In simulation, we evaluate new tasks such as **separating-piles**.
> |            | single task+0     | Single task + 2    | Single task +4    |
> |------------|------------|------------|------------|
> | Average    | 29.7 | 40.1| 49.3 |
>
> As shown in the above table, the experiment results are also consistent. The GPT-generated tasks demonstrate efficacy across a more diverse array of objects.
>  In the real world, we have also evaluated novel tasks with assets such as fruits and spoons.
> |            |   organize-kits  | sort-garbage  | place-bread   |
> |------------|------------|------------|------------|
> | Average    | 60.0 | 40.0 | 30.0 |
>
> We have updated Figure 8 and Table 1 in the main paper to reflect the new experiments, we showed that training and evaluating on more diverse tasks such as **spoons and piles** are also possible, while the main bottleneck is the hardware. Given our time budget, we showed that the policies pre-trained from 70 GenSim tasks can outperform policies pre-trained from 10 CLIPort tasks and training from scratch. Due to the limited time and hardware limitations (only suction cup), we chose these tasks (see [website](gen-sim.github.io) for visualizations), and we believe our pipeline and pretrained weights can be useful for more tasks.
>
> There are two reasons behind our task selection: task-level generalization and sim-to-real transfer.
>
> 1. We highlight that we did not focus our evaluation on diverse assets such as rope, because we focus more on *task-level generalization*. We consider “build-a-car” to also be a challenging task that involves. We note that in *Table 2*, we have already shown performance on tasks on “move-piles-along-line” and “manipulating-two-ropes”. We believe that when scaling is improved, we will see more interesting generalizations to both diverse tasks and scenes due to the robustness of ``common-sense`` representations.
>
> 2. We point out that our experiment focuses on pretraining on large simulation data and transferring it to the real world. For CLIPort, it only focused on sim-to-sim and real-to-real for demonstration purposes. In fact, in Table 1, similar zero-shot sim-to-real baselines will have 0 success rates due to the perception gaps. However, we are interested in evaluating novel-task generalization in simulation and conducting sim-to-real by using policies bootstrapped from the simulation. We also use a different hardware than the CLIPort (we use xARM with a suction cup), so dexterous tasks with very complex geometry in the real world can be difficult.
>
> > Experiments presented in Fig 6 left) evaluate models on the same tasks as CLIPort but it is not clear if the models were pre-trained on all generated tasks in this experiment (see W3.)
>
> Thanks for asking. We would like to know clarifications on the Figures. It looks like that you might be referring to *Figure 7* since *Figure 6* does not include CLIPort experiments. We do not pre-train on the evaluation tasks or pre-train on any tasks in this few-shot generalization experiment. Moreover, the GPT-generated training tasks are selected based on task code embeddings produced by LLM model “text-embedding-ada-002”. Please feel free to clarify the question and let us know if this needs a follow-up.
>
> > W3. The “few-shot policy generalization to related tasks” experiment needs additional description to understand the meaning of “single-original, single+2, single+4 etc”. Were the models pre-trained on original clipport tasks, original clipport tasks + 2 generated tasks etc? Or are the models pre-trained on all CLIPort tasks + all the ones that were generated automatically? It is important to clarify this experiment.
>
> Thanks for your suggestion.  We have added additional descriptions for Figure 7 and the few-shot policy generalization section in the main paper. The x-axis denotes single CLIPort task + N GPT tasks where N=0,2,4 and multiple CLIPort tasks + N GPT tasks where N=0,4,8. We use the GPT embedding to pick related tasks from the evaluation tasks. The goal of the experiment is to see if the additional related tasks and data generated by GPT-4 can be used to improve the training performance. We clarify that the models are not pre-trained on any tasks. This experiment does not have large-scale pre-training, we train few-shots on CLIPort (single/multiple)+N tasks from GenSim and evaluate the CLIPort tasks.

---

> ### Author Response · Authors · 2023-11-19
> **Thank you for your constructive feedback! (3)**
>
> > W4. Page 6 mentions “When only pretraining on the 10 tasks in CLIPort, the policy does not generalize well on the GPT4 tasks.” In fig 7 right) the model pretrained on tasks generated with LLMs is better than the one trained on CLipport tasks, but the difference is 15-20% of relative improvements, the claim should therefore be mitigated. It is also suggested to justify how the set of testing tasks (given in Appendix A.1) was selected. See also W2.
>
> We acknowledge that the claim on the performance of pre-training on CLIPort tasks should be mitigated and we have removed that sentence. However, we would like to point to the additional results, which state that pre-trained CLIPort tasks indeed only have limited zero-shot generalization in our simulation experiments (only 1.6% success rate among 4 GPT generated tasks). We also note that the right subfigure in Figure 7 is few-shot joint training instead of zero-shot generalization. The tasks in Appendix A.1 are used for evaluating LLM task generation across different LLMs (from GPT-3 finetuned to GPT-4). The set of testing tasks is not particularly selected.
>
>
> > W5. Details of the metrics used in Fig 6 would be appreciated. In particular, how is “code reasoning capability” measured in “runtime-verified”, and is the “task completed” an average over several environment resets ?
>
> There are two levels of mistakes that LLMs make. The metric mostly measures low-level mistakes. Our metric has an incremental structure from syntax to runtime to successfully generate demonstrations where failing the former metrics implies failing the latter metrics. “Runtime verified” measures if the code can be runnable without any runtime problems such as out of bound, “Task completed” is computed as 1 if we have 3 trials managing to generate demonstrations over 5 trials (sometimes there could be simulation issues). Therefore, they test whether the LLM can successfully write the code to generate demonstrations. Our additional human experiments also measure that the human pass rate is around 0.5. We also note that evaluating LLM judiciously is still an open research question.
>
> I am however concerned about the diversity of the tasks which the method is evaluated on. In particular, the evaluation tasks seem to be less varied compared to the ones used in CLIPort (W2). I am also questioning the “automatic” nature of the method as human intervention is required (W1). I will gladly update my review if the authors can address these concerns and respond to my questions (see weaknesses).
> We have conducted additional experiments for these questions (please also check the main paper). We summarize our answers to these questions below and please see our full answers to W1-W5 above:
> Diversity of task evaluation:
> We have conducted additional experiments to verify that our framework can also be extended to more varied tasks as suggested. In the simulation, we evaluate new tasks such as "separating-piles". In the real world, we have also evaluated novel tasks with assets such as fruits and spoons. Please see our detailed answers with experiments to W2. Our additional experiment shows that the policy can also generalize to diverse tasks and assets.
>
> For task selection rationale, we emphasize that the evaluation tasks are selected to focus on sim-to-real transfer and task-level generalization. For simulation, we choose interesting long-horizon tasks such as build-a-car over diverse assets such as rope and kits in both task generation and policy evaluations. However, we note that we have evaluated on manipulating-two-ropes and and move-piles-along line in table 2. For the real world, we are interested in sim-to-real transfer (in contrast to sim-to-sim and real-to-real in CLIPort) and are limited by the real-world hardware (suction cup) and thus focus on tasks involving simple structures. Finally, we want to highlight that we are still limited by compute budgets, where each experiment can take a week.
>
> Automatic nature and human effort: please see our general response and response to your W1.
>
> Thank you very much again for your constructive feedback and insightful questions. We appreciate any follow-up questions or comments.

---

> ### Comment · Reviewer_1vRz · 2023-11-20
>
> Thank you for the detailed explanations and additional experiments.
>
> My concerns regarding:
> - The human efforts required to annotate the generated tasks.
> - The selection of tasks for the experiments in Fig 7.
> - The variety of tasks.
>
> have been addressed. I now better understand that the paper focuses on task generalization across long-horizon tasks, and that the real experiments are limited by the available hardware. Thanks also for pointing to the experiments in Table 2 which indeed already evaluate the method for rope manipulation tasks.
>
> Several clarifications have been also added to the paper. Finally, I suggest to clarify the following sentence in the paper.
> > The set of testing tasks is not particularly selected.
>
> Following the authors response, I have increased the rating of my review and recommend to accept this paper.

---

> > ### Author Response · Authors · 2023-11-20
> > **Thanks again for the suggestions!**
> >
> > Dear Reviewer, Thank you very much for the suggestions! We have added the clarification to the paper.

---

### Official Review · Reviewer_XopG · 2023-11-01

**Soundness:** 4 excellent
**Presentation:** 4 excellent
**Contribution:** 4 excellent
**Rating:** 8
**Confidence:** 4

**Summary:**

This paper proposes a methodology to automatically generate tasks in simulation and corresponding expert policies to increase the task diversity when training robot policies. Existing large language models are utilized for this, and a number of ablations are conducted to study various design choices of the proposed method. Experimental evaluation both in simulation and in the real world reveals that the proposed methodology leads to better generalization than the baseline.

**Strengths:**

- Interesting idea. The proposed idea is very interesting and novel.
- Well-written. The paper is very well-written and easy to follow. The figures very much aid in understanding the paper.
- Experimental Evaluation. The experiments reveal that the proposed methodology outperforms baseline methods by non-trivial margins. In particular, the proposed method enables both few-shot policy generalization to related tasks and zero-shot policy generalization to unseen tasks. The experiments are conducted in both simulation and in the real world, which further speak to the strength of the proposed method.
- Ablations. Ablations are conducted to better understand various design choices of the proposed methodology. Visualizations and qualitative results also aid in this.

**Weaknesses:**

- Simplistic tasks. The experimentation is limited to table-top pick-and-place task. Furthermore, the use of a suction gripper is a weakness -- this simplifies the already simple task, and limits generalizability to other kinds of tasks. Unclear whether the proposed methodology can be adapted for more complex tasks, and how well it would do. Experiments on more complex tasks would be more convincing.
- Experimental Evaluation. While a diverse set of experiments are conducted, only one setting (ie environment) is adopted. The results would be more convincing if the presented trends were shown to hold true across a number of settings.

**Questions:**

- A number of approaches exist in the literature for data augmentation using foundation models (e.g. ROSIE, CACTI, etc.). How do such approaches compare to the proposed methodology when it comes to generalization to unseen tasks? Arguably, some of these are simpler to implement and use.
- What is done in the cases where the output of the LLM has a syntactical error and cannot be run?
- How much prompt engineering went into this?

---

> ### Author Response · Authors · 2023-11-19
> **Thank you for your constructive feedback!**
>
> Thank you for finding the topic of our work is of increasing importance, and appreciating our contribution. We have revised the manuscript and will address your questions below.
>
> > Simplistic tasks and Experimental Evaluation. The experimentation is limited to table-top pick-and-place task.While a diverse set of experiments are conducted, only one setting (ie environment) is adopted. The results would be more convincing if the presented trends were shown to hold true across a number of settings.
>
> While the domain of top-down pick-and-place, especially short-horizon tasks, are *seemingly* not challenging to most roboticists, we note that the  **task-level policy generalization** explored in GenSim, such as long-horizon tasks and compositional tasks, remains under-explored and non-trivial. For example, "build-a-car" is a nontrivial task that requires semantic understanding of natural language instructions and more than 10 steps of precise pick-and-place and stacking. Such task-level generation cannot be achieved by previous methods without LLMs, as future work we plan to extend our work to handle more challenging tasks such as dexterous, 6-dof, contact-rich tasks.
>
>
> > A number of approaches exist in the literature for data augmentation using foundation models (e.g. ROSIE, CACTI, etc.). How do such approaches compare to the proposed methodology when it comes to generalization to unseen tasks? Arguably, some of these are simpler to implement and use.
>
> Thanks for bringing up these interesting works; we will include these references in the related work section. From our understanding, ROSIE and CACTI focus more on **scene-level generalization** on a single task, utilizing image diffusions for data augmentation. However, in our work, we focus on **task-level** generalization with LLM for generating more tasks. As these works are orthogonal to our approach, it would be interesting to explore a combination of both to achieve generalization across unseen tasks in diverse scenes.
>
> >  What is done in the cases where the output of the LLM has a syntactical error and cannot be run?
>
> One of the properties of coding simulation tasks is that we can just execute the robot simulation and check the results. Therefore, if the output of LLM is non-executable, we will do **rejection sampling** and move to the next trial of LLM generation. We have added a new Appendix Section D.5 on the feedback in GenSim. Feel free to check that section out.
>
> > How much prompt engineering went into this?
>
> During the LLM pipeline development process, we went through prompt engineering, retrievals and reflections, and finetuning and used techniques such as chain-of-thoughts and few-shot prompting. The final version of the prompt is *not* very complicated and contains two subtasks of generating the task descriptions and the task code (Figure 3). Figure 6 Left shows the necessity of this subtask separation in prompt chains to give LLM enough “time” to think. The prompts are attached in Appendix Section D.2. One intuition is that creating simulation task descriptions and implementing the code for the tasks require different language capabilities. For creating the task, we would like to generate tasks as creative and interesting as possible (with high temperature for decoding) while being physically feasible, so it tests the language reasoning capabilities of LLM. For implementing the task code in simulation, it requires rigorous coding execution (with 0 temperature for decoding) and needs to be thoughtful in the details of syntax and goal generation. In summary, the two subtask pipeline is important and this can avoid complicated prompt engineering.
>
> Thank you very much again for your constructive feedback. Please do not hesitate to let us know if you have any other questions and/or comments.

---

> > ### Author Response · Authors · 2023-11-22
> > **Thank you for your constructive feedback!**
> >
> > Dear reviewer, thanks so much for the response, and hope our additional clarification and experiments have addressed the issues. If possible, we wonder if you would kindly consider raising the scores. Thanks tremendously again for the helpful suggestions for improving the paper!

---

### Author Response · Authors · 2023-11-19
**General Response to All Reviewers (1)**

We thank all the reviewers for their valuable and constructive comments. We have revised the paper as suggested (highlighted in blue) by the reviewers, and have addressed their comments. Overall, we appreciate that most reviewers acknowledged the contributions and significance of our work as “very interesting and novel” (from Reviewer XopG), addressing “an important problem and the method is technically sound” (from Reviewer 1vRz) and noted the tasks are “inspiring and impressive” (from Reviewer hwME) with “rigorous and practical impacts” (from Reviewer LF4g). We address several common questions from the reviewers below.

**Clarification on Tasks for Evaluation**

While the domain of top-down pick-and-place, especially short-horizon tasks, is *seemingly* not challenging to most roboticists, we note that the  **task-level policy generalization** explored in GenSim, such as long-horizon tasks and compositional tasks, remains under-explored and non-trivial. For example, "build-a-car" is a nontrivial task that requires a semantic understanding of natural language instructions and more than 10 steps of precise pick-and-place and stacking. Despite the underlying primitives are still pick-and-place, generalization across these tasks are difficult and such task-level generation cannot be achieved by previous methods without LLMs.


**New Experiments:**

We have updated the writing and figures (Figure 7, Figure 8, Table 1) based on the comments on the experiment sections. To clarify the task selection in joint training. We use task code embedding via LLM model `text-embedding-ada-002`  to find nearby tasks for joint training. For evaluation tasks, we have conducted more simulation and real-world experiments to demonstrate that the findings in policy training are not tied to a particular selection of simple tasks (such as blocks).
We detail the specific experiments below:

Real-world Experiment on More Tasks: We have conducted more experiments on 3 new real-world tasks, sort-garbage, organize-kits, and place-bread. Our success rate across three tasks is 43.3%, a 16.7% improvement compared to the baseline pre-trained only with CLIPort tasks and 10% improvement over models without pretrain. Please check our [website](gen-sim.github.io) for visualizations of the rebuttal experiments. One of the reasons for failure cases is due to hardware limitations: suction cups can fail when handling complex geometry without a smooth surface.
|            |  No adaptation    |  No pretrain    | CLIPort    | GenSim (50 Tasks) | GenSim (70 Tasks) |
|------------|------------|------------|------------|------------|------------|
|  organize-kits    | 0 | 50  | 50 | 30 | 60 |
| sort-garbage   | 0  | 30 | 10  | 40  | 40 |
| place-bread    | 0  | 20 | 20 | 10 | 30  |
| Average    | 0  | 33.3 | 26.7 | 26.7 | 43.3 |

Multiple Sets of Train/Test Tasks in Simulation: We first note that Table 2 already includes performance on tasks such as "manipulate-two-ropes" and "sweep-piles". We have experimented with another set of test tasks (**separating-piles**) in the joint training experiments in Figure 7 left. We have similar observations on the improvement of few-shot generalization by including additional GenSim tasks. We also added below joint training performance with GenSim tasks in the few-shot settings. We have also added more zero-shot generalization results in Figure 7 middle.
|            | Single task+0     | Single task + 2    | Single task +4    | Multi task+0     | Multi task + 4    | Multi task + 8    |
|------------|------------|------------|------------|------------|------------|------------|
| Average    | 29.7 | 40.1 | 49.3| 2.8 | 31 | 36.5|

---

### Author Response · Authors · 2023-11-19
**General Response to All Reviewers (2)**

**Human Feedback for Task Selection:**

Regarding the “automatic” nature of the pipeline, we note that the amount of human effort is indeed needed, but it is drastically reduced compared with creating simulation tasks from scratch.  First, 90% of the error cases can be filtered out via automatic metrics such as execution and environment feedback. Second, the amount of human effort per task for annotation is also greatly reduced. In particular, the human annotator only needs to check correctness (i.e., success or not) based on the rendered video of simulation tasks — we have quantitatively analyzed the human effort by re-generating 30 tasks during the rebuttal process and measuring human effort in terms of wall-clock time, please see Appendix D.6). Overall, we found that without human annotation (i.e., completely automatic), most generated tasks are already useful qualitatively for task-level generalization. For future work, we think that scaling up human feedback (e.g., using Amazon Turk) is one of the key considerations for further scaling up the generation of high-quality tasks.


We detail the specific experiment below:

LLM Evaluation Experiment on Measuring Human Effort: We generated 30 new tasks and measured the time spent and pass rates given by three users with different backgrounds in robotic simulation. We highlight that this process requires only coarse feedback and typically takes less than 10 seconds to look at the generated videos and associated text instructions of the tasks. We show our findings below.
| Task       | User 1 Time | User 1 Pass | User 2 Time |User 2  Pass | User 3 Time |User 3 Pass |
|------------|------------|------------|------------|------------|------------|------------|
| Average    | 12.83 | 0.8 | 5.11  | 0.63 | 17.4 | 0.41 |



**Writing:**

We have edited our manuscript to improve the clarity of the experiment and figures (highlighted in blue). Specifically, we have included explicit discussions of specific failure cases as well as human efforts in *Appendix Section D.4, D.5, D.6, D.7* to discuss detailed error messages, feedback, human efforts, and scalability.  Finally, we wish to reemphasize our focus on *task-level generalization* rather than *asset diversity*. We plan to release a language-conditioned benchmark (along with model weights) generated by GPT for the study of scaling policy learning parametrized with affordance prediction. This will include task sets and pre-trained models from GensSim-5 to GenSim-100 for future research.

---

### Meta-Review · Area_Chair_Yq6S · 2023-12-05

**Metareview:**

This work proposes GenSim, an approach to automatically generate robotics tasks using LLMs. This line of work has only recently been possible due to recent advances in LLMs, and offers a path to generating sufficiently diverse data for training embodied agents. The results clearly demonstrate the improvement by generating larger numbers of tasks, which can motivate further scaling in the community to get more general agents. Furthermore, the code is going to be open sourced so it can also be used as a benchmark for future LLMs, since we clearly see a gap between GPT4 and the next best models. This means that the impact of the paper is magnified as it is a promising future direction. The main weaknesses are that the work is currently only applied to a single tabletop domain and there is still the requirement for a human in the loop, but these can be addressed in future work.

**Justification For Why Not Higher Score:**

The main issues are 1) it is limited to just tabletop domains. 2) It still requires human in the loop to verify the tasks, rather than for example using some kind of automatic detection.

**Justification For Why Not Lower Score:**

This work is worthy of a spotlight because it is timely, as one of the first papers to generate diverse tasks using LLMs and showing it can subsequently improve the learned policies for sim2real transfer.

---

### Decision · Program_Chairs · 2024-01-16

Accept (spotlight)